# What's a good imputation to predict with missing values?

**Marine Le Morvan**[1,2]   **Julie Josse**[4]   **Erwan Scornet**[3]   **Gaël Varoquaux**[1]

[1] Université Paris-Saclay, Inria, CEA, Palaiseau, 91120, France
[2] Université Paris-Saclay, CNRS/IN2P3, IJCLab, 91405 Orsay, France
[3] CMAP, UMR7641, Ecole Polytechnique, IP Paris, 91128 Palaiseau, France
[4] Inria Sophia-Antipolis, Montpellier, France

{marine.le-morvan, julie.josse, gael.varoquaux}@inria.fr
erwan.scornet@polytechnique.edu

## Abstract

How to learn a good predictor on data with missing values? Most efforts focus on first imputing as well as possible and second learning on the completed data to predict the outcome. Yet, this widespread practice has no theoretical grounding. Here we show that for almost all imputation functions, an impute-then-regress procedure with a powerful learner is Bayes optimal. This result holds for all missing-values mechanisms, in contrast with the classic statistical results that require missing-at-random settings to use imputation in probabilistic modeling. Moreover, it implies that perfect conditional imputation is not needed for good prediction asymptotically. In fact, we show that on perfectly imputed data the best regression function will generally be discontinuous, which makes it hard to learn. Crafting instead the imputation so as to leave the regression function unchanged simply shifts the problem to learning discontinuous imputations. Rather, we suggest that it is easier to learn imputation and regression jointly. We propose such a procedure, adapting NeuMiss, a neural network capturing the conditional links across observed and unobserved variables whatever the missing-value pattern. Experiments confirm that joint imputation and regression through NeuMiss is better than various two step procedures in our experiments with finite number of samples.

## 1 Introduction

Data with missing values are ubiquitous in many applications, as in health or business: some observations come with missing features. There is a rich statistical literature on imputation as well as inference with missing values [Rubin, 1976, Little and Rubin, 1987, 2002, 2019]. Most of the theory and practices build upon the *Missing At Random* (MAR) assumption that allows to maximize the likelihood of observed data while ignoring the missing-values mechanism, for instance using expectation maximization [Dempster et al., 1977]. On the contrary, Missing Not At Random settings, where missingness depends on the unobserved values, may not be identifiable and require dedicated methods with a model of the missing-values mechanism.

Learning predictive models with missing values poses distinct challenges compared to inference tasks [Josse et al., 2019]. Indeed, when the input is an arbitrary subset of variables in dimension $d$, there are $2^d$ potential missing data patterns and as many sub-models to learn. Consequently, even simple data-generating mechanisms lead to complex decision rules [Le Morvan et al., 2020b]. To date, there are few supervised-learning models natively suited for partially-observed data. A notable

35th Conference on Neural Information Processing Systems (NeurIPS 2021).

exception is found with tree-based models [Twala et al., 2008, Chen and Guestrin, 2016], widely used in data-science practice.

The most common practice however remains by far to use off-the-shelf methods first for imputation of missing values and second for supervised-learning on the resulting completed data. Such a procedure may benefit from progress in missing-value imputation with machine learning [van Buuren 2018, Yoon et al. 2018, Mattei and Frellsen 2019]. However, there is a lack of learning theory to support such Impute-then-Regress procedures: Under what conditions are they Bayes consistent? Which aspects of the imputation are important?

There is empirical realization that the choice of imputation matters for predictive performance. The NADIA R package [Borowski and Fic] can select an imputation method to minimize a prediction error on a test set. Auto-ML is used to optimize full pipelines, including imputation [eg Jarrett et al., 2021]. Ipsen et al. [2020] optimize a constant imputation for supervised learning. However, the imputation is only weakly guided by the target in these approaches, it is set either from a family of black-box methods using gradient-free model selection, or from trivial imputation functions. In addition, there is a lack of insight on what drives a good imputation for prediction.

We contribute a systematic analysis of Impute-the-Regress procedures in a general setting: non-linear response function and any missingness mechanism (no MAR assumptions). We show that:

- Impute-then-Regress procedures are Bayes optimal for *all missing data mechanisms* and for *almost all imputation functions*, whatever the number of variables that may be missing. This very general result gives theoretical grounding to such widespread procedures.

- We study "natural" choices of imputation and regression functions: the oracle imputation by the conditional expectation and oracle regression function on the complete data. We show that chaining these oracles is not Bayes optimal in general and quantify its excess risk. We show that in both cases, choosing an oracle for one step, imputation or regression, imposes discontinuities on the other step, thus making it harder to learn.

- As these results suggest that imputation and regression should be adapted to one another, we contribute a method that jointly optimizes imputation and regression, using NeuMiss networks [Le Morvan et al., 2020a] as a differentiable imputation procedure.

- We compare empirically a number of Impute-then-Regress procedures on simulated non-linear regression tasks. Joint optimization of both steps provides the best performance.

## 2 Problem setting

**Notations**  We consider a dataset of i.i.d. realizations of the random variable $(X, M, Y) \in \mathbb{R}^d \times \{0, 1\}^d \times \mathbb{R}$ where $X$ are the complete covariates, $M$ a missingness indicator, and $Y$ a response of interest. For each realization $(x, m, y)$, $m_j = 1$ indicates that $x_j$ is missing, and $m_j = 0$ that it is observed. We denote by $mis(m) \subset [\![1, d]\!]$ the indices corresponding to the missing covariates (and similarly $obs(m)$ the observed indices), so that $x_{obs(m)}$ corresponds to the entries actually observed. We define the incomplete covariate vector $\widetilde{X} \in (\mathbb{R} \cup (\texttt{NA}))^d$ as $\widetilde{X}_j = X_j$ if $M_j = 0$ and $\widetilde{X}_j = \texttt{NA}$ otherwise, where $\texttt{NA}$ represents a missing value.

**Assumptions**  We assume that $X$ admits a density on $\mathbb{R}^d$ and that, for all $j \in [\![1, d]\!]$, each component $X_j$ has finite expectation and variance, that is $\mathbb{E}\left[X_j^2\right] < \infty$. Moreover, we assume that the response $Y$ is generated according to:

$$Y = f^\star(X) + \epsilon, \qquad \text{with } \mathbb{E}\left[\epsilon | X_{obs(M)}, M\right] = 0 \quad \text{and } \mathbb{E}\left[Y^2\right] < \infty. \tag{1}$$

where $f^\star : \mathbb{R}^d \to \mathbb{R}$ is a function of the complete input data $X$, $\epsilon \in \mathbb{R}$ is a random noise variable.

### 2.1 Supervised learning with missing values

**Optimization problem**  In practice, in the presence of missing values, we do not have access to the complete data $(X, M, Y)$ but only to the subset of it that is observed, i.e, $(X_{obs(M)}, M, Y)$. Thus instead of learning a mapping from $\mathbb{R}^d$ to $\mathbb{R}$, we need to learn a mapping from $(\mathbb{R} \cup (\texttt{NA}))^d$ to $\mathbb{R}$, where the set of observed covariates can be any subset of $[\![1, d]\!]$. It is this unusual input space, partly discrete, that makes supervised learning with missing values challenging and different from classical

supervised learning problems. Formally, the optimization problem we wish to solve is:

$$\min_{f:(\mathbb{R}\cup(\text{NA}))^d \mapsto \mathbb{R}} \mathcal{R}(f) := \mathbb{E}\left[\left(Y - f(\widetilde{X})\right)^2\right] \tag{2}$$

**Bayes predictor**    The function which minimizes (2), called the *Bayes predictor*, is given by:

$$\tilde{f}^\star(\widetilde{X}) = \mathbb{E}\left[Y|X_{obs(M)}, M\right] = \mathbb{E}\left[f^\star(X)|X_{obs(M)}, M\right]. \tag{3}$$

As $\widetilde{X}$ is a function of $X_{obs}$ and $M$, we will sometimes slightly abuse notations and write $\tilde{f}^\star(\widetilde{X}) = \tilde{f}^\star(X_{obs}, M)$. The risk of the Bayes predictor is called the *Bayes risk*, which we denote as $\mathcal{R}^\star$. It is the lowest achievable risk for a given supervised learning problem.

**Definition 1** (Bayes optimality). *A Bayes optimal function $f$ achieves the Bayes rate, i.e, $\mathcal{R}(f) = \mathcal{R}^\star$.*

As can be seen from (3), the Bayes predictor is a function of $M$, a discrete random variable that can take one of $2^d$ values since $M \in \{0, 1\}^d$. The function $\tilde{f}^\star$ can thus be viewed as $2^d$ different functions, one for each possible subset of variables. This view raises questions that are central to this paper: How should we parametrize functions on such input domains? And which function families should we consider to approximate $\tilde{f}^\star$? These questions have been studied in the case where $f^\star$ is assumed to be a linear function, and $X$ follows a Gaussian distribution. Indeed, under these assumptions, Le Morvan et al. [2020b,a] have derived analytical expressions for the Bayes predictor and deduced appropriate parametric estimators. However, aside from specific cases, it is impossible to derive an analytical expression for the Bayes predictor, and novel arguments are needed to understand which classes of functions should be considered in general.

## 3    Asymptotic analysis of Impute-then-regress procedures

### 3.1    Impute-then-regress procedures

Let $|mis(m)|$ (resp. $|obs(m)|$) be the number of missing entries (resp. observed) for any missing data pattern $m$. For each $m \in \{0, 1\}^d$, we define an *imputation function* $\phi^{(m)} : \mathbb{R}^{|obs(m)|} \to \mathbb{R}^{|mis(m)|}$ which outputs values for the missing entries based on the observed ones. We denote by $\phi_j^{(m)} : \mathbb{R}^{|obs(m)|} \to \mathbb{R}$ the component function of $\phi^{(m)}$ that imputes the $j$-th component in $X$ if it is missing. Classical choices of imputation functions include constant functions or linear functions. Finally, we introduce the family of functions $\mathcal{F}^I$ that transform an incomplete vector into a complete one, precisely:

$$\mathcal{F}^I = \left\{\Phi : (\mathbb{R} \cup \{\text{NA}\})^d \to \mathbb{R}^d : \forall j \in [\![1, d]\!], \, \Phi_j(\widetilde{X}) = \begin{cases} X_j & \text{if } M_j = 0 \\ \phi_j^{(M)}(X_{obs(M)}) & \text{if } M_j = 1 \end{cases}\right\}. \tag{4}$$

Let us define $\mathcal{F}_\infty^I$ in the exact same way but for imputation functions $\phi^{(m)} \in \mathcal{C}^\infty$, for all $m \in \{0, 1\}^d$. Here we study *Impute-then-regress procedures*, which we define as two-step procedures where the data is first imputed using a function $\Phi \in \mathcal{F}^I$, and then a regression is performed on the imputed data. Such a procedure is quite natural to deal with arbitrary subsets of inputs variables. It embeds the data into $\mathbb{R}^d$ to reduce the problem to a classical one. In practice, impute-then-regress procedures are widely used. However, the choice of function class is so far mostly ad-hoc and raises multiple questions: How close to the Bayes rate can functions obtained via such procedures be? Should we prefer some choices of imputation functions over others? What happens when the missing data mechanism is missing not at random? In this section, we will give answers to these questions.

Below, we write $obs$ (resp. $mis$) instead of $obs(M)$ (resp. $mis(M)$) to lighten notations.

### 3.2    Impute-then-regress procedures are Bayes optimal

**Definition 2** (Universal consistency). *An estimator $f_n$ is Bayes consistent if $\lim_{n\to\infty} \mathcal{R}(f_n) = \mathcal{R}^\star$. It is said to be universally consistent if the previous statement holds for all distributions of $(X, Y)$.*

The following theorem shows that Impute-then-regress procedures are Bayes optimal for almost all imputation functions. In other words, it means that a universal learner trained on imputed data provides optimal performances asymptotically for almost all imputation functions. Let us now define, for all imputation functions $\Phi \in \mathcal{F}^I$, the function $g_\Phi^\star \in \underset{g:\mathbb{R}^d \mapsto \mathbb{R}}{\mathrm{argmin}} \quad \mathbb{E}\left[\left(Y - g \circ \Phi(\widetilde{X})\right)^2\right].$

**Theorem 3.1** (Bayes consistency of Impute-then-regress procedures). *Assume the data is generated according to* (1). *Then, for almost all imputation function* $\Phi \in \mathcal{F}_\infty^I$, *the function* $g_\Phi^\star \circ \Phi$ *is Bayes optimal. In other words, for almost all imputation functions* $\Phi \in \mathcal{F}_\infty^I$, *a universally consistent algorithm trained on the imputed data* $\Phi(\widetilde{X})$ *is Bayes consistent.*

Appendix A.3 gives the proof. Theorem 3.1 states a very general result: Impute-then-regress procedures are Bayes consistent for all missing data mechanisms, almost all imputation functions, regardless of the distribution of $(X, Y)$ and the number of missing covariates. Since Theorem 3.1 holds for almost all imputation functions, it implies that good imputations are not required to obtain good predictive performances, at least asymptotically. Note that here, the notion of *almost all* is to be understood in its topological sense, and not in its measure theory sense. Moreover, this theorem does not make any assumption on the missing data mechanism, and is therefore valid for Missing Not At Random (MNAR) data. This contrasts with most methods for inference and imputation with missing values, valid only for MAR data. Finally, the theorem remains valid for any configuration of variables that may contain missing values, including the case in which all variables may contain missing values. Bayes consistency of Impute-the-Regress procedures has already been studied, but in much more restricted settings. Josse et al. [2019] proved that such procedures are Bayes consistent under the MAR assumption, for constant imputations functions and for only one potentially missing variable. Bertsimas et al. [2021] refined this result to almost surely continuous imputation functions. While these two prior works build on very similar proofs, we use here very different arguments summarized in the next paragraph.

The first key idea of the proof is that, after imputation, all data points with a given missing data pattern $m$ are mapped to a manifold $\mathcal{M}^{(m)}$ of dimension $|obs(m)|$. For example in 3D, data points are mapped to $\mathbb{R}^3$ when completely observed, to 2D manifolds when they have one value missing, to 1D manifolds when they have two values missing, and to one point when all values are missing (see Figure 1). Thus, Impute-then-Regress procedures first map data points to various manifolds depending on their missing data patterns and then apply a prediction function defined on the whole space including manifolds. The second key idea of the proof is to ensure that the original missing data patterns of imputed points can almost surely be identified. For this, the proof requires that all manifolds of the same dimension are pairwise transverse. This assumption is sufficient, though not necessary, to ensure that the intersection of two manifolds of dimension $|obs(m)|$ cannot itself be of dimension $|obs(m)|$. Transversality is a weak assumption. In fact, Thom's transversality theorem, (which we rely on in our proof) says that it is a generic property: it holds for "typical examples", i.e *almost all* imputation functions will lead to transverse manifolds. To clarify this concept, we provide a particular case in 2D where 1D manifolds are not transverse in Appendix A.4.

The proof is constructive and exhibits a function $g_\Phi^\star$ which achieves the Bayes rate for a given set of imputation functions. For each manifold $\mathcal{M}^{(m)}$, ordered from smallest dimension to largest, we require that $g_\Phi^\star$ on $\mathcal{M}^{(m)}$ equals the Bayes predictor for missing data pattern $m$ except on points for which $g_\Phi^\star$ has already been defined, i.e, the points where $\mathcal{M}^{(m)}$ intersects with the manifolds ranked before it. Thus, we obtain a function $g_\Phi^\star$ that does not depend on $m$, and which for each manifold, equals the Bayes predictor except on subsets of measure zero under the assumption that manifolds of the same dimension are pairwise transverse. Refer to appendix A.3 for more details.

Figure 1: **Example - Imputation manifolds in three dimensions** — 3-dimensional Gaussian data after imputation. Data points are colored according to their missing data pattern prior to imputation. Red, brown and purple (resp. orange, blue, and green) correspond to missing data patterns with two (resp. one) missing value(s). Completely observed points are not represented to ease the visualization of manifolds.

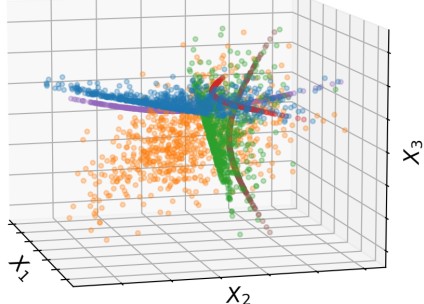

While this theorem is a very general result, it does not say what the optimal function associated to a given imputation looks like. In fact, depending on the imputation function it may be non-continuous, vary widely, and require a very large number of samples to be approximated.

**Note on Impute-then-classify procedures -** Theorem 3.1 applies to regression problems. However, it can easily be shown that a similar result holds in binary classification settings. Indeed, in a binary classification setting, the Bayes predictor predicts class 1 if $P(Y = 1|X) > 0.5$ and -1 otherwise. Thus, it suffices to consider that the function of interest $f^\star(X)$ is the posterior probability $P(Y = 1|X)$. Then the same arguments as those used to prove Theorem 3.1 can be used to show that Impute-then-classify procedures are Bayes optimal for almost all imputation functions.

# 4 Imputation versus regression: choosing one may break the other

Theorem 3.1 gives a theoretical grounding to Impute-then-regress procedures. As it holds for almost any imputation function, one could very well choose simple and cheap imputations such as imputing by a constant. However, the difficulty of the ensuing learning problem will depend on the choice of imputation function. Indeed, the function $g_\Phi^\star$ that achieves the Bayes rate depends on the imputation function $\Phi$. In general, it may not be continuous or smooth. Thus $g_\Phi^\star$ can be more or less difficult to approximate by machine learning algorithms depending on the chosen imputation function.

Le Morvan et al. [2020b] showed that even if $Y$ is a linear function of $X$, imputing by a constant leads to a complicated Bayes predictor: piecewise affine but with $2^d$ regions. This result highlights how imputations neglecting the structure of covariates can result in additional complexity for the regression function $g_\Phi^\star$. Rather, another common practice is to impute by the conditional expectation: it minimizes the mean squared error between the imputed matrix and the complete one and is the target of most imputation methods. One hope may be that if $f^\star$ has desirable properties, such as smoothness, conditional imputation will lead to a function $g_\Phi^\star$ which inherits from these properties.

In this section we first show that replacing missing values by their conditional expectation in the oracle regression function $f^\star$ gives a small but non-zero risk. Characterizing the optimal function on the conditionally-imputed data, we find that it suffers from discontinuities and thus forms a difficult estimation problem. Rather, we study whether the imputation can be corrected for $f^\star$ to form the Bayes predictor on partially-observed data.

## 4.1 Applying $f^\star$ on conditional imputations: chaining oracles isn't without risks.

The conditional imputation function $\Phi^{CI} : (\mathbb{R} \cup \{\texttt{NA}\})^d \to \mathbb{R}^d$ is defined as follows:

$$\forall j \in [\![1, d]\!], \; \Phi_j^{CI}(\widetilde{X}) = \begin{cases} X_j & \text{if } M_j = 0 \\ \mathbb{E}[X_j|X_{obs}, M] & \text{if } M_j = 1 \end{cases}$$

Note that $\Phi^{CI} \in \mathcal{F}^I$. To lighten notations, we will write $X^{CI} := \Phi^{CI}(\widetilde{X})$ to denote the conditionally imputed data.

**Lemma 4.1** (First order approximation). *Assume that the data is generated according to (1). Moreover assume that (i) $f^\star \in \mathcal{C}^2(\mathcal{S}, \mathbb{R})$ where $\mathcal{S} \subset \mathbb{R}^d$ is the support of the data, and that (ii) there exists positive semidefinite matrices $\bar{H}^+ \in P_d^+$ and $\bar{H}^- \in P_d^+$ such that for all $X$ in $\mathcal{S}$, $\bar{H}^- \preccurlyeq H(X) \preccurlyeq \bar{H}^+$ with $H(X)$ the Hessian of $f^\star$ at $X$. Then for all $X$ in $\mathcal{S}$ and for all missing data patterns:*

$$\frac{1}{2} tr \left( \bar{H}_{mis,mis}^- \Sigma_{mis|obs,M} \right) \leq \tilde{f}^\star(\widetilde{X}) - f^\star(X^{CI}) \leq \frac{1}{2} tr \left( \bar{H}_{mis,mis}^+ \Sigma_{mis|obs,M} \right) \tag{5}$$

*where $\Sigma_{mis|obs,M}$ is the covariance matrix of the distribution of $X_{mis}$ given $X_{obs}$ and $M$.*

Appendix A.6 gives the proof. The assumption that $\bar{H}^- \preccurlyeq H(X) \preccurlyeq \bar{H}^+$ for any $X$ means that the minimum and maximum curvatures of $f^\star$ in any direction are uniformly bounded over the entire space. Lemma 4.1 shows that applying $f^\star$ to the conditionally imputed (CI) data is a good approximation of the Bayes predictor when there is no direction in which both the curvature of $f^\star$ and the conditional variance of the missing data given the observed one are high. Intuitively, if a low quality imputation is compensated by a flat function, or conversely, if a fast varying function is compensated by a high quality imputation, then $f^\star$ applied to the CI data approximates well the Bayes predictor.

**Proposition 4.1** ((Non-)Consistency of chaining oracles). *The function $f^\star \circ \Phi^{CI}$ is Bayes optimal if and only if the function $f^\star$ and the imputed data $X^{CI}$ satisfy:*

$$\forall M \text{ s.t. } P(M) > 0, \quad \mathbb{E}\left[f^\star(X)|X_{obs}, M\right] = f^\star(X^{CI}) \quad \text{almost everywhere.} \tag{6}$$

*Besides, under the assumptions of Lemma 4.1, the excess risk of chaining oracles compared to the Bayes risk $\mathcal{R}^\star$ is upper-bounded by:*

$$\mathcal{R}(f^\star \circ \Phi^{CI}) - \mathcal{R}^\star \leq \frac{1}{4}\mathbb{E}_M\left[\max\left(tr\left(\bar{H}_{mis,mis}^- \Sigma_{mis|obs,M}\right)^2, tr\left(\bar{H}_{mis,mis}^+ \Sigma_{mis|obs,M}\right)^2\right)\right] \tag{7}$$

Appendix A.7 gives the proof. Condition (6) for Bayes optimality is clearly stringent. Indeed, if one variable is missing, condition (6) says that the expectation of the regression function should be equal to the regression function applied at the expected entry. Although such functions are difficult to characterize precisely, it is clear that condition (6) is difficult to fulfill for generic regression functions (linear functions are among the few examples that do satisfy it). Therefore, for most functions $f^\star$, $f^\star \circ \Phi^{CI}$ is not Bayes optimal. Proposition 4.1 also gives an upper bound for the excess risk of the predictor $f^\star(X^{CI})$ compared to the Bayes rate, showing here again that if there is no direction in which both the curvature and the variance of the missing data given the observed one are high, the excess risk is small.

*The special case of linear regression:* When $f^\star$ is a linear function, the curvature is 0, hence eq. (7) implies no excess risk. This is also visible from the expression of the Bayes predictor (3), where the expectation on unobserved data can be pushed inside $f^\star$ as it is linear. The Bayes predictor can thus be exactly written as $f^\star$ applied to conditionally-imputed data.

### 4.2 Regressing on conditional imputations, a good idea?

**Proposition 4.2** (Regression function discontinuities). *Suppose that $f^\star \circ \Phi^{CI}$ is not Bayes optimal, and that the probability of observing all variables is strictly positive, i.e., for all $x$, $P(M = (0, \ldots, 0), X = x) > 0$. Then there is no continuous function $g$ such that $g \circ \Phi^{CI}$ is Bayes optimal.*

In other words, when conditional imputation is used, the optimal regression function experiences discontinuities unless it is $f^\star$. The proof is given in appendix A.8. From a finite-sample learning standpoint, discontinuous functions are in general harder to learn: in the general case, non-parametric regression requires more samples to achieve a given error on functions without specific regularities as opposed to functions with a form of smoothness [see e.g., Györfi et al., 2006, chap 3]. Hence, while regression on conditional imputation may be consistent (Theorem 3.1), it can require an inordinate number of samples.

### 4.3 Fasten your seat belt: corrected imputations may experience discontinuities.

Another possible route is to find *corrected imputations* which we define as imputation functions $\Phi$ such that, if $f^\star$ is used as regression function, the impute-then-regress procedure $f^\star \circ \Phi$ is Bayes optimal. Intuitively, given a fixed regression function $f^\star$, the question is: can we "correct" an imputation function and thus the manifold that it describes so that $f^\star$ restricted to this manifold is equal to the Bayes predictor? Assuming $f^\star$ is continuous, the intermediate value theorem gives a first answer to this question by ensuring the existence of imputations functions satisfying

$$f^\star \circ \Phi(X_{obs(M)}, M) = \mathbb{E}\left[f^\star(X)|X_{obs(M)}, M\right].$$

For the same reasons as above, determining that such imputations not only exist but are *continuous* is important from a practical perspective. Indeed, assuming $f^\star$ is continuous, the Bayes predictor with missing values could then be tackled as the composition of two continuous functions, with an Impute-then-Regress strategy. Intuitively in 2D, the existence of a continuous corrected imputation can be seen as the existence of a continuous path in the 2D plane whose value by $f^\star$ equals the Bayes predictor. Figure 2 (left) gives a simple example in 2D for which a continuous corrected imputation exists. Here if one chooses the imputation function of $X_2$ given $X_1$ as the black function denoted as $\Phi_{corrected}$, then its composition by the green paraboloid $f^\star$ gives the Bayes Predictor depicted on the right in black. By contrast, if one imputes by the conditional expectation, then its composition with $f^\star$ gives the red curve which is different from the Bayes Predictor. Note that the manifolds in Figure 1 were obtained using (continuous) corrected imputations functions for the same setting as Figure 2 (left) but with 3-dimensional data. However, as illustrated in Figure 2 (right), *continuous* corrected

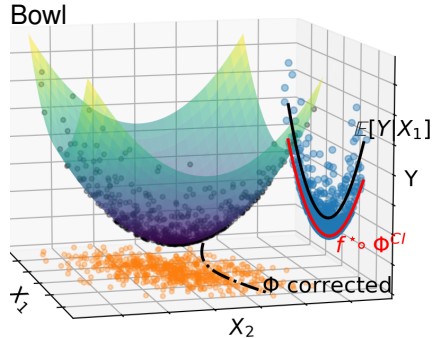
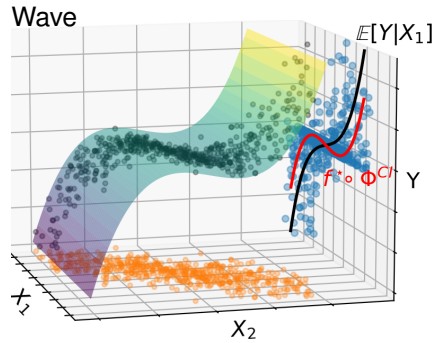

Figure 2: **Left: corrected imputation** The regression function is $f^\star(x_1, x_2) \mapsto x_1^2 + x_2^2$. When $x_2$ is missing, chaining perfect conditional imputation with the regression function ($f^\star \circ \Phi^{CI}$) gives a biased predictor, shown in red, as the unexplained variance in $x_2$ is turned into bias. However, using as an imputation $\Phi(x_1) = \sqrt{\rho^2 x_1^2 + (1 - \rho^2)}$ corrects this bias, with $\rho$ the correlation between $x_1$ and $x_2$. **Right: no continuous corrected imputation exists**. The function is defined as $f^\star(x_1, x_2) \mapsto x_2^2 - 3 x_2$. No continuous corrected imputation is possible because the Bayes predictor on the partially-observed data $\mathbb{E}[Y|X_1]$ is monotonous, while the regression function $f^\star$ is not.

imputations do not always exist. Indeed, on this example the Bayes predictor is non-decreasing but there is no continuous path in the 2D plane on which $f^\star$ is non-decreasing and maps at some point to both the 'purple' and 'yellow values' (proof in Appendix A.9). It is thus of interest to clarify when continuous corrected imputations exist. Proposition 4.3 establishes such conditions.

**Proposition 4.3** (Existence of continuous corrected imputations). *Assume that $f^\star$ is uniformly continuous, twice continuously differentiable and that, for all missing patterns $m$ and all $x_{obs}$, the support of $X_{mis}|X_{obs} = x_{obs}, M = m$ is connected. Additionally, assume that for all missing patterns $m$, and all $(x_{obs}, x_{mis})$, the gradient of $f^\star$ with respect to the missing coordinates is nonzero:*

$$\nabla_{x_{mis}} f^\star(x_{obs}, x_{mis}) \neq 0. \tag{8}$$

*Then, for all $m$, theres exist continuous imputation functions $\phi^{(m)} : \mathbb{R}^{|obs(m)|} \to \mathbb{R}^{|mis(m)|}$ such that $f^\star \circ \Phi$ is Bayes optimal.*

Appendix A.10 gives a proof based on a global implicit function theorem. Assumption 8 is restrictive: it is for instance not met for our example in Figure 2 (left), which still admits continuous corrected imputations. This highlights the fact that continuous corrected imputations also exist under weaker conditions, but it is difficult to conclude on "how often" it is the case.

## 5 Jointly optimizing an impute-n-regress procedure: NeuMiss+MLP

The above suggests that it is beneficial to adapt the regression function to the imputation procedure and vice versa. Hence, we introduce a method for the joint optimization of these two steps by chaining a NeuMiss network with an MLP (multi-layer perceptron).

NeuMiss [Le Morvan et al., 2020a] is a neural-network architecture originally designed to approximate the Bayes predictor for linear models with missing values. It contains a Neumann block that has the particularity of using element-wise multiplications by the missingness indicator as non-linearities. Here we reuse this block to play the role of an imputation layer. This choice is motivated by two key reasons. First, the Neumann block is a theoretically grounded layer for missing values: it can approximate the conditional expectation of the missing values given the observed ones with an error that decays exponentially fast with its depth. As explained in Prop. 4.1, this property is desirable in some cases. Second, it is a differentiable block, which allows it to be chained with a MLP and learned jointly with the regression function. The resulting architecture can thus be seen as an Impute-then-Regress architecture, but that can be jointly optimized.

We performed one minor improvement on the NeuMiss architecture compared to the original paper. Though the theory behind NeuMiss points to using shared weights in the Neumann block as well as residual connections going from the input to each hidden layer of the Neumann block, Le Morvan et al. [2020a] used neither. We found empirically that shared weights in the Neumann block as well

as residual connections improved performance. Therefore, we used both in all our experiments. For clarity, the (non-linear) NeuMiss architecture is described in detail in Appendix B.1.

# 6 Empirical study of impute-n-regress procedures

## 6.1 Experimental setup

**Data generation** The data $X \in \mathbb{R}^{n \times d}$ are generated according to a multivariate Gaussian distribution $\mathcal{N}(\mu, \Sigma)$ where the mean is drawn from a standard Gaussian and the covariance is generated as $\Sigma = BB^\top + D$. $B \in \mathbb{R}^{d \times q}$ is a matrix with entries drawn from a standard normal Gaussian distribution, and $D$ is a diagonal matrix with small entries that ensures that the covariance matrix is full rank. We study two correlation settings called *high* and *low* corresponding respectively to $q = \texttt{int}(0.3 * d)$ and $q = \texttt{int}(0.7 * d)$. The experiments are run with $d = 50$.

**Choice of $f^\star$** The response $Y$ is generated according to $Y = f^\star(X) + \epsilon$ with three choices of $f^\star$ named *bowl*, *wave*, and *break* depicted in Figure 3 (exact expression in appendix B.2). $\beta$ is a vector of ones normalized such that the quantity $z = \beta^\top X + \beta_0$ follows a Gaussian distribution centered on 1 with variance 1. Note that $f^\star_{bowl}$, $f^\star_{wave}$ and $f^\star_{break}$ were designed so that the desired variations occur over the support of the data. The noise $\epsilon$ is chosen so as to have a signal-to-noise ratio of 10.

Figure 3: **Bowl, wave and break functions** used for $f^\star$ in the empirical study.

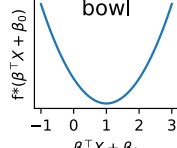
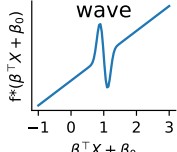
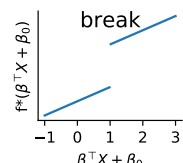

**Missing values** 50% of the entries of $X$ were deleted according to one of two missing data mechanisms: Missing Completely At Random (MCAR) or Gaussian self-masking [GSM, see Le Morvan et al., 2020a]. Gaussian self-masking is a Missing Not At Random (MNAR) mechanism, where the probability that a variable $j$ is missing depends on $X_j$ via a Gaussian function.

**Baseline methods benchmarked** For each level of correlation (*low* or *high*), for each function $f^\star$ (*bowl*, *wave* or *break*), and each missing data mechanism (MCAR or GSM), we compare a number of methods. First, for reference, we compute various oracle predictors:

- **Bayes predictor**: This is the function that achieves the lowest achievable risk. In general cases, its expression cannot be derived analytically. However, we show that it can be derived for ridge functions, i.e. functions of the form $x \mapsto g(\beta^\top x)$, for some choices of $g$ including polynomials, the Gaussian cdf and piecewise constant functions. We thus built $f^\star_{bowl}$, $f^\star_{wave}$ and $f_{break}$ as combination of these base functions which allows us to compute their corresponding Bayes predictors. Appendix B.2 gives their expressions.
- **Chained oracles**: $f^\star \circ \Phi^{CI}$ consists in imputing by the conditional expectation and then applying $f^\star$. The analytical expression of $\Phi^{CI}$ can be derived analytically for both MCAR and GSM, and we thus use this analytical expression to impute the missing values.
- **Oracle + MLP**: The data is imputed using the analytical expression of the conditional expectation, and then a MLP is fitted to the completed data.

These three predictors all use ground truth information (parameters $\mu$, $\Sigma$ of the data distribution, expression of $f^\star$ or of the missing data mechanism) which are unavailable in practice. They are mainly useful as reference points. We then compare the NeurMiss+MLP architecture and a number of classic Impute-then-Regress methods as well as gradient boosted regression trees:

- **Mean + MLP** The data is imputed by the mean, and a multilayer perceptron (MLP) is fitted to the completed data.
- **MICE + MLP** The data is imputed using Scikit-learn's [Pedregosa et al., 2012, BSD licensed] conditional imputer `IterativeImputer` that adapts the popular Multivariate Imputation by Chained Equations [MICE, van Buuren, 2018] to be able to impute a test set. A multilayer perceptron (MLP) is then fitted to the completed data.

- **GBRT**: Gradient boosted regression trees (Scikit-learn's `HistGradientBoostingRegressor` with default parameters). This predictor readily supports missing values: during training, missing values on the decision variable for a given split are sent to the left or right child depending on which provides the largest gain. This is know as the Missing Incorporated Attribute strategy [Twala et al., 2008].

Finally, we also run **Mean + mask + MLP** as well as **MICE + mask + MLP** in which the mask is concatenated to the imputed data before fitting a MLP. Concatenating the mask is a widespread pratice to account for MNAR data.

All MLPs are implemented with PyTorch [Paszke et al., 2019]. A validation set is used to choose MLPs' depth (1, 2 or 5), width ($1d$, $5d$ or $10d$), initial learning rate (ranging from $5.10^{-4}$ to $10^{-2}$) and weight decay (ranging from $10^{-6}$ to $10^{-3}$). Adam is used with an adaptive learning rate: the learning rate is divided by 5 each time 10 consecutive epochs fail to decrease the training loss by at least 1e-4. Early stopping is triggered when the validation score does not improve by at least 1e-4 for 12 consecutive epochs. The batch size is set to 100, and ReLUs are used as activation functions. Finally for NeuMiss the depth is set to 20. Note that since the weights of NeuMiss are shared, increasing its depth does not increase its number of parameters. For gradient boosted regression trees, several hyperparameters are chosen using the validation set including the maximum number of leaves for each tree (from 50 to 600), the maximum number of iterations for the boosting process (from 100 to 300), as well as the minimum number of samples per leaf (from 10 to 50).

The experiments use training sets of size $n = 100\,000$ and validation and test sets of size $n = 10\,000$. The code for all experiments is available at `https://github.com/marineLM/Impute_then_Regress`.

## 6.2 Experimental results

The results are presented in Figure 4 as well as in Figure 8 (Appendix B.3).

**Chaining oracles fails when both curvature is high and correlation is low.** The chained oracle has a performance close to that of the Bayes predictor in all cases except when the wave or break functions are applied to low correlation data. This observation illustrates well Proposition 4.1. Intuitively, the Bayes predictor for each missing data pattern is a smoothed version of $f^\star$, and it is all the more smoothed that there is uncertainty around the likely values of the missing data. In the

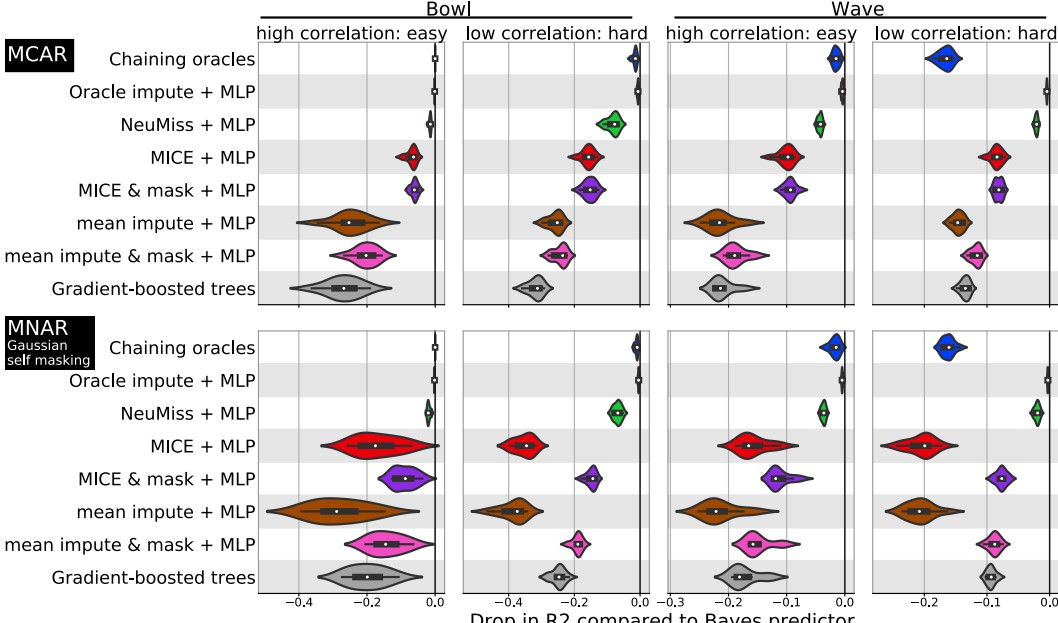

Figure 4: Performances (R2 score on a test set) compared to that of the Bayes predictor across 10 repeated experiments.

low correlation setting, the uncertainty is such that $f^\star$ is not a good proxy anymore for the Bayes predictor.

**Regressing on oracle conditional imputation provide excellent performances.**   Contrary to the chained oracles, *Oracle + MLP* is close to the Bayes rate in all cases. This result should be put into perspective with Proposition 4.2, which states that there is no *continuous* regression function $g$ such that $g \circ \Phi^{CI}$ is Bayes optimal unless it is $f^\star$. Indeed, as the MLP can only learn continuous functions, it shows that there are continuous functions $g$ such that $g \circ \Phi^{CI}$, even though it is not Bayes optimal, performs very well.

**Adding the mask is critical in MNAR settings with *mean* and *MICE* imputations**   In MNAR settings, missingness carries information that can be useful for prediction. However, both the mean and iterative conditional imputation output an imputed dataset in which the missingness information is more difficult to retrieve. For this reason, it is common practice to concatenate the mask with the imputed data to expose the missingness information to the predictor. Our experiments show that under self-masking (MNAR), adding the mask to the mean or iteratively imputed data markedly improves performances. Note that NeuMiss does not require adding the mask as an input since the missingness information is already incorporated via the non-linearities.

**NeuMiss+MLP performs best among Impute-then-Regress predictors.**   In *all* settings, Neu-Miss performs best. GBRT performs poorly here possibly because they are not well adapted to approximate smooth functions. Finally, note that when the difficulty of the problem increases, for example with a lower correlation, then (i) the performance of the Bayes predictor decreases and (ii) the differences in performance among methods is reduced, as in the lower right panel.

## 7   Conclusion

Impute-then-regress procedures assemble standard statistical routines to build predictors suited for data with missing values. However, we have shown that seeking the best prediction of the outcome leads to different tradeoffs compared to inferential purposes. Given a powerful learner, *almost all imputations* lead asymptotically to the optimal prediction, *whatever the missingness mechanism*. A good choice of imputation can however reduce the complexity of the function to learn. Though conditional expectation can lead to discontinuous optimal regression functions, our experiments show that it still leads to easier learning problems compared to simpler imputations. In order to adapt the imputation to the regression function, we proposed to jointly learn these two steps by chaining a trainable imputation via the NeuMiss networks and a classical MLP. An empirical study of non-linear regression shows that it outperforms impute-then-regress procedures built on standard imputation methods as well as gradient-boosted trees with incorporated handling of missing values. In further work, it would be useful to theoretically characterize the learning behaviors of Impute-then-Regress methods in finite sample regimes.

### Acknowledgments and Disclosure of Funding

MLM, JJ, and GV acknowledge funding via DataIA MissingBigData. MlM and GV acknowledge funding via ANR-17-CE23-0018 DirtyData and GV acknowledges funding via ANR-20-CHIA-0026 LearnI. JJ acknowledges funding via ANR-16-IDEX-0006.

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
