# Supplementary materials – What's a good imputation to predict with missing values?

## A Proofs

### A.1 Proof of Lemma A.1

**Lemma A.1.** *Let $\phi^{(m)} \in \mathcal{C}^\infty \left( \mathbb{R}^{|obs(m)|}, \mathbb{R}^{|mis(m)|} \right)$ be the imputation function for missing data pattern $m$, and let $\mathcal{M}^{(m)} = \left\{ x \in \mathbb{R}^d : x_{mis} = \phi^{(m)}(x_{obs}) \right\}$. For all $m$, $\mathcal{M}^{(m)}$ is an $|obs|-$dimensional manifold.*

*Proof.* Let:

$$h^{(m)} : \mathbb{R}^d \to \mathbb{R}^{|mis|}$$

$$x \mapsto x_{mis} - \phi^{(m)}(x_{obs})$$

Regular value: We will show that $\mathbf{0}_{mis}$ is a regular value of $h^{(m)}$. By definition [see p21 in Guillemin and Pollack, 1974], a point $y \in \mathbb{R}^{|mis|}$ is a regular value of $h^{(m)}$ if $dh_x^{(m)}$ is surjective at every point $x$ such that $h^{(m)}(x) = y$. The mapping $dh_x^{(m)}$ is linear and can be represented by the Jacobian of $h^{(m)}$ at $x$:

$$J_{h^{(m)}}(x) = \left( \begin{array}{c|c} A & Id \end{array} \right), \quad A \in \mathbb{R}^{|mis| \times |obs|}, Id \in \mathbb{R}^{|mis| \times |mis|}.$$

Given the structure of $J_{h^{(m)}}(x)$, it is obviously of rank $|mis|$ at every point $x$. Thus $dh_x^{(m)}$ is surjective at every point $x$, and it is true in particular for the points $x$ such that $h^{(m)}(x) = \mathbf{0}$. We conclude that by definition, $\mathbf{0}_{mis}$ is a regular value of $h^{(m)}$.

Preimage theorem: By the Preimage theorem ([Guillemin and Pollack, 1974], p.21), since $\mathbf{0} \in \mathbb{R}_{mis}$ is a regular value of $h^{(m)} : \mathbb{R}^d \to \mathbb{R}^{|mis|}$, then the the preimage $\left( h^{(m)} \right)^{-1}(\mathbf{0})$ is a submanifold of $\mathbb{R}^d$ of dimension $d - |mis| = |obs|$.

Since by definition, $\left( h^{(m)} \right)^{-1}(\mathbf{0}) = \mathcal{M}^{(m)}$, we have that $\mathcal{M}^{(m)}$ is a $|obs|-$dimensional mainfold.
□

### A.2 Proof of Lemma A.2

**Lemma A.2.** *Let $m$ and $m'$ be two distinct missing data patterns with the same number of missing values $|mis|$. Let $\phi^{(m)} \in \mathcal{C}^\infty \left( \mathbb{R}^{|obs(m)|}, \mathbb{R}^{|mis(m)|} \right)$ be the imputation function for missing data pattern $m$, and let $\mathcal{M}^{(m)} = \left\{ x \in \mathbb{R}^d : x_{mis} = \phi^{(m)}(x_{obs}) \right\}$. We define similarly $\phi^{(m')}$ and $\mathcal{M}^{(m')}$. For almost all imputation functions $\phi^{(m)}$ and $\phi^{(m')}$,*

$$dim \left( \mathcal{M}^{(m)} \cap \mathcal{M}^{(m')} \right) = \begin{cases} 0 & if \ |mis| > \frac{d}{2} \\ d - 2|mis| & otherwise. \end{cases} \tag{9}$$

*Proof.* According to Thom Transversality theorem ([Golubitsky, 1973], p.54) with:

- $W = \mathcal{M}^{(m')}$,
- $f = \phi^{(m)}$,
- $k = 0$ (note that as stated p.37, $J^0(X, Y) = X \times Y$ and $j^0 f(x) = \text{graph}(f)$),

we have that $\left\{ \phi^{(m)} \in \mathcal{C}^\infty(\mathbb{R}^{|obs|}, \mathbb{R}^{|mis|}) \mid \mathrm{graph}(\phi^{(m)}) \pitchfork \mathcal{M}^{(m')} \right\}$ is a residual subset of $\mathcal{C}^\infty(\mathbb{R}^{|obs|}, \mathbb{R}^{|mis|})$ in the $\mathcal{C}^\infty$ topology. In other words, the fact that $\mathrm{graph}(\phi^{(m)})$ is transverse to $\mathcal{M}^{(m')}$ is a generic property. Put differently, almost all functions $\phi^{(m)}$ have their graph transverse to $\mathcal{M}^{(m')}$. Note that here the notion of *almost all* has to be understood in its topological sense, and not in its measure theory sense.

Suppose that $|obs| < \frac{d}{2}$. According to Lemma A.1, $\mathcal{M}^{(m')}$ is a $|obs|$−dimensional manifold. Moreover we just showed that for almost all $\phi^{(m)}$, $\mathrm{graph}(\phi^{(m)}) \pitchfork \mathcal{M}^{(m')}$. Applying Proposition 4.2 of [Golubitsky, 1973] (p.51) with $W = \mathcal{M}^{(m')}$ and $f = \mathrm{graph}(\phi^{(m)})$, we obtain that $\mathcal{M}^{(m)}$ and $\mathcal{M}^{(m')}$ are disjoint, since, by definition, $\mathcal{M}^{(m)}$ is the image of $\mathrm{graph}(\phi^{(m)})$. Consequently, the dimension of their intersection is 0.

Suppose that $|obs| \geq \frac{d}{2}$. According to the theorem p.30 of [Guillemin and Pollack, 1974], since $\mathcal{M}^{(m)}$ and $\mathcal{M}^{(m')}$ are transverse submanifolds of $\mathbb{R}^d$, their intersection is again a manifold with $\mathrm{codim}(\mathcal{M}^{(m)} \cap \mathcal{M}^{(m')}) = \mathrm{codim}(\mathcal{M}^{(m)}) + \mathrm{codim}(\mathcal{M}^{(m')})$. This implies that $\dim(\mathcal{M}^{(m)} \cap \mathcal{M}^{(m')}) = 2|obs| - d$. $\qquad\qquad\square$

## A.3 Proof of Theorem 3.1

**Theorem 3.1** (Bayes consistency of Impute-then-regress procedures). *Assume the data is generated according to* (1). *Then, for almost all imputation function $\Phi \in \mathcal{F}_\infty^I$, the function $g_\Phi^\star \circ \Phi$ is Bayes optimal. In other words, for almost all imputation functions $\Phi \in \mathcal{F}_\infty^I$, a universally consistent algorithm trained on the imputed data $\Phi(\widetilde{X})$ is Bayes consistent.*

*Proof.* Let $\phi^{(m)} \in \mathcal{C}^\infty\left(\mathbb{R}^{|obs(m)|}, \mathbb{R}^{|mis(m)|}\right)$ be the imputation function for missing data pattern $m$, and let $\mathcal{M}^{(m)} = \left\{ x \in \mathbb{R}^d : x_{mis} = \phi^{(m)}(x_{obs}) \right\}$. According to Lemma A.1, for all $m$, $\mathcal{M}^{(m)}$ is an $|obs|$−dimensional manifold. $\mathcal{M}^{(m)}$ corresponds to the subspace where all points with missing data pattern $m$ are mapped after imputation.

Let us order missing data patterns according to their number of missing values, with the pattern of all missing entries ranked first and the pattern of all observed entries ranked last. Two patterns with the same number of missing values are ordered arbitrarily. We use $m(i)$ to refer to the missing data pattern ranked in $i^{th}$ position.

Let $g^\star$ be the function defined as follows: for all $i$,

$$\forall Z = \Phi(\widetilde{X}) \in \mathcal{M}^{(m(i))} \setminus \bigcup_{m(k) < m(i)} \mathcal{M}^{(m(k))}, \qquad g^\star(Z) = \tilde{f}^\star(\widetilde{X}).$$

For a given missing data pattern $m(i)$, by distributivity of intersections across unions, we have:

$$\mathcal{M}^{(m(i))} \bigcap \left( \bigcup_{m(k) < m(i)} \mathcal{M}^{(m(k))} \right) = \bigcup_{m(k) < m(i)} \left( \mathcal{M}^{(m(i))} \bigcap \mathcal{M}^{(m(k))} \right)$$

If $m(k)$ has strictly more missing values than $m(i)$, then by Lemma A.1 $\dim(\mathcal{M}^{(m(k))}) < \dim(\mathcal{M}^{(m(i))})$, and thus $\dim(\mathcal{M}^{(m(k))} \cap \mathcal{M}^{(m(i))}) < \dim(\mathcal{M}^{(m(i))})$. Moreover, If $m(k)$ has the same number of missing values as $m(i)$, then by Lemma A.2, for almost all imputation functions $\phi^{(m(k))}$ and $\phi^{(m(i))}$, $\dim(\mathcal{M}^{(m(k))} \cap \mathcal{M}^{(m(i))}) < \dim(\mathcal{M}^{(m(i))})$. We conclude that for all $m(k) < m(i)$, $\mathcal{M}^{(m(k))} \cap \mathcal{M}^{(m(i))}$ is a subset of measure zero in $\mathcal{M}^{(m(i))}$. Finally, since a countable union of sets of measure zero has measure zero, we obtain that $\bigcup_{m(k) < m(i)} \left( \mathcal{M}^{(m(i))} \cap \mathcal{M}^{(m(k))} \right)$ has measure zero in $\mathcal{M}^{(m(i))}$.

Let's now compute the risk of $g^\star \circ \Phi$:

$$\mathcal{R}(g^\star \circ \Phi) = \sum_{M=m} P(M = m) \int_{X_{obs}} P(X_{obs}|M = m) \left( \tilde{f}^\star(\widetilde{X}) - g^\star \circ \Phi(\widetilde{X}) \right)^2 \qquad (10)$$

For a given missing data pattern $m$, $\Phi(\widetilde{X}) \in \mathcal{M}^{(m)}$. Moreover, we constructed $g^\star$ such that $g^\star \circ \Phi(\widetilde{X}) = \tilde{f}^\star(\widetilde{X})$ for all $\Phi(\widetilde{X}) \in \mathcal{M}^{(m)}$ except on a set that we just showed to be of measure zero for almost all imputation functions. As a result, the function $X_{obs} \mapsto \tilde{f}^\star(\widetilde{X}) - g^\star \circ \Phi(\widetilde{X})$ is zero almost everywhere for a given $m$, and the function $X_{obs} \mapsto P(X_{obs}|M = m)\left(\tilde{f}^\star(\widetilde{X}) - g^\star \circ \Phi(\widetilde{X})\right)^2$ is also zero almost everywhere. Since the integral of a function that vanishes almost everywhere is equal to 0, we conclude that $\mathcal{R}(g^\star \circ \Phi) = 0$. Since the risk cannot be negative, $g^\star \circ \Phi$ is a minimizer of the risk and thus it is Bayes optimal. $\qquad\square$

## A.4  Examples of transverse and nontransverse manifolds in 2D.

Theorem 3.1 is true for *almost all* imputation functions and not *all* of them. Thus, we can construct examples with particular choices of imputation functions that lead to nontransverse manifolds, and consequently for which Impute-then-Regress procedures are not Bayes optimal. We provide such an example below.

Consider a dataset with points $x \in \mathbb{R}^2$, and let $a \in \mathbb{R}$. Let $\Phi_2^{(0,1)}(x_1) = a * x_1$ be the imputation function for $x_2$ when only $x_1$ is observed. And let $\Phi_1^{(1,0)}(x_2) = \frac{1}{a}x_2$ be the imputation function for $x_1$ when only $x_2$ is observed. In this particular case shown in Figure 5 (bottom), the manifolds on which the data with either $x_1$ missing or $x_2$ missing are projected are exactly the same (the same line in the 2D space). Thus they are nontransverse and consequently Theorem 3.1 does not hold.

However according to the Thom transversality theorem, almost all imputation functions will lead to transverse manifolds.

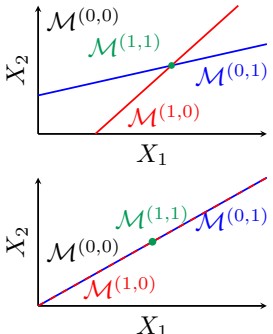

Figure 5: **Example - Linear imputation manifolds in two dimensions** Manifolds represented for linear imputation functions. $\mathcal{M}^{(0,0)}$ is the whole plane. Note that $\mathcal{M}^{(1,1)}$ need not be at the intersection of the two lines, it depends on the imputation function chosen. With linear imputation functions, $\mathcal{M}^{(0,1)}$ and $\mathcal{M}^{(1,0)}$ are transverse if and only if the two lines are not coincident.**Top:** Transverse manifolds. **Bottom:** Nontransverse manifolds.

## A.5  Proof of Lemma A.3

**Lemma A.3.**
$$\forall X \in \mathbb{R}^p, \ \forall mis \subseteq [\![1,p]\!], \ H(X) \preccurlyeq \bar{H}^+ \implies H_{mis,mis}(X) \preccurlyeq \bar{H}^+_{mis,mis}$$

*Proof.* Let $X \in \mathbb{R}^p$, and let $m$ be a missing data pattern with observed (resp. missing) indices $obs$ (resp. $mis$). $H(X) \preccurlyeq \bar{H}^+$ is equivalent to:

$$\forall u \in \mathbb{R}^p, \ u^\top \left(\bar{H}^+ - H(X)\right) u \geq 0. \tag{11}$$

Let $\mathcal{V} \subseteq \mathbb{R}^p$ be a subspace such that for any $v$ in $\mathcal{V}$, $v_{obs} = 0$. Since $\mathcal{V} \subseteq \mathbb{R}^p$, (11) implies:

$$\forall v \in \mathcal{V}, \ v^\top \left(\bar{H}^+ - H(X)\right) v \geq 0$$
$$\iff \forall v_{mis} \in \mathbb{R}^{|mis|}, \ v_{mis}^\top \left(\bar{H}^+_{mis,mis} - H_{mis,mis}(X)\right) v_{mis} \geq 0$$
$$\iff H_{mis,mis}(X) \preccurlyeq \bar{H}^+_{mis,mis}$$

$\qquad\square$

## A.6  Proof of Lemma 4.1

**Lemma 4.1** (First order approximation)**.** *Assume that the data is generated according to (1). Moreover assume that (i) $f^\star \in \mathcal{C}^2(\mathcal{S}, \mathbb{R})$ where $\mathcal{S} \subset \mathbb{R}^d$ is the support of the data, and that (ii)*

*there exists positive semidefinite matrices $\bar{H}^+ \in P_d^+$ and $\bar{H}^- \in P_d^+$ such that for all $X$ in $\mathcal{S}$, $\bar{H}^- \preccurlyeq H(X) \preccurlyeq \bar{H}^+$ with $H(X)$ the Hessian of $f^\star$ at $X$. Then for all $X$ in $\mathcal{S}$ and for all missing data patterns:*

$$\frac{1}{2}tr\left(\bar{H}^-_{mis,mis}\Sigma_{mis|obs,M}\right) \leq \tilde{f}^\star(\widetilde{X}) - f^\star(X^{CI}) \leq \frac{1}{2}tr\left(\bar{H}^+_{mis,mis}\Sigma_{mis|obs,M}\right) \tag{5}$$

*where $\Sigma_{mis|obs,M}$ is the covariance matrix of the distribution of $X_{mis}$ given $X_{obs}$ and $M$.*

*Proof.* Without loss of generality, suppose that we reorder variables such that we can write $X = (X_{obs}, X_{mis})$. Consider the function

$$f^\star_{mis} : \mathbb{R}^{|mis|} \to \mathbb{R}$$
$$X_{mis} \mapsto f^\star(X_{obs}, X_{mis})$$

Since $f^\star \in \mathcal{C}^2\left(\mathbb{R}^d, \mathbb{R}\right)$, we have $f^\star_{mis} \in \mathcal{C}^2\left(\mathbb{R}^{|mis|}, \mathbb{R}\right)$. Therefore, we can write the first order Taylor expansion (see Theorem 2.68 in Folland [2002]) of $f^\star_{mis}$ around $E\left[X_{mis}|X_{obs}, M\right]$:

$$\begin{aligned}
f^\star_{mis}(X_{mis}) =& f^\star(X_{obs}, \mathbb{E}\left[X_{mis}|X_{obs}, M\right]) \\
&+ \nabla f^\star_{mis}(X_{obs}, \mathbb{E}\left[X_{mis}|X_{obs}, M\right])^\top (X_{mis} - \mathbb{E}\left[X_{mis}|X_{obs}, M\right]) \\
&+ R\left(X_{mis} - \mathbb{E}\left[X_{mis}|X_{obs}, M\right]\right),
\end{aligned} \tag{12}$$

where $R$ is the Lagrange remainder satisfying

$$R\left(X_{mis} - \mathbb{E}\left[X_{mis}|X_{obs}, M\right]\right) =$$
$$\frac{1}{2}\left(X_{mis} - \mathbb{E}\left[X_{mis}|X_{obs}, M\right]\right)^\top H_{mis,mis}(c)\left(X_{mis} - \mathbb{E}\left[X_{mis}|X_{obs}, M\right]\right),$$

for some $c$ in the ball $\mathcal{B}\left(\mathbb{E}\left[X_{mis}|X_{obs}, M\right], \| X_{mis} - \mathbb{E}\left[X_{mis}|X_{obs}, M\right] \|_2\right)$. By assumption, for all $X$, $H(X) \preccurlyeq \bar{H}^+$. Therefore, according to Lemma A.3, we have $H_{mis,mis}(X) \preccurlyeq \bar{H}^+_{mis,mis}$ for any missing data pattern, which leads to:

$$R\left(X_{mis} - \mathbb{E}\left[X_{mis}|X_{obs}, M\right]\right) \leq$$
$$\frac{1}{2}\left(X_{mis} - \mathbb{E}\left[X_{mis}|X_{obs}, M\right]\right)^\top \bar{H}^+_{mis,mis}\left(X_{mis} - \mathbb{E}\left[X_{mis}|X_{obs}, M\right]\right).$$

Using equality (12), we get:

$$\begin{aligned}
f^\star(X_{obs}, &X_{mis}) - f^\star(X_{obs}, \mathbb{E}\left[X_{mis}|X_{obs}, M\right]) \\
&- \nabla f^\star_{mis}(X_{obs}, \mathbb{E}\left[X_{mis}|X_{obs}, M\right])^\top (X_{mis} - \mathbb{E}\left[X_{mis}|X_{obs}, M\right]) \\
&\leq \frac{1}{2}\left(X_{mis} - \mathbb{E}\left[X_{mis}|X_{obs}, M\right]\right)^\top \bar{H}^+_{mis,mis}\left(X_{mis} - \mathbb{E}\left[X_{mis}|X_{obs}, M\right]\right)
\end{aligned}$$

Finally, taking the expectation with regards to $P(X_{mis}|X_{obs}, M)$ on both sides, we obtain

$$\mathbb{E}\left[f^\star(X_{obs}, X_{mis})|X_{obs}, M\right] - f^\star(X_{obs}, \mathbb{E}\left[X_{mis}|X_{obs}, M\right]) \leq \frac{1}{2}tr(H^{+\top}_{mis,mis}\Sigma_{mis|obs,M}), \tag{13}$$

where we have used the fact that, for any vector $X \in \mathbb{R}^d$ and for any $H \in P_d^+$,

$$X^\top H X = tr(X^\top H X) = tr(HXX^\top).$$

Following a similar reasoning, we can show that:

$$\mathbb{E}\left[f^\star(X_{obs}, X_{mis})|X_{obs}, M\right] - f^\star(X_{obs}, \mathbb{E}\left[X_{mis}|X_{obs}, M\right]) \geq \frac{1}{2}tr(H^{-\top}_{mis,mis}\Sigma_{mis|obs,M}) \tag{14}$$

Together, inequalities (13) and (14) conclude the proof. $\qquad\square$

## A.7 Proof of Proposition 4.1

**Proposition 4.1** ((Non-)Consistency of chaining oracles). *The function $f^\star \circ \Phi^{CI}$ is Bayes optimal if and only if the function $f^\star$ and the imputed data $X^{CI}$ satisfy:*

$$\forall M \text{ s.t. } P(M) > 0, \quad \mathbb{E}\left[f^\star(X)|X_{obs}, M\right] = f^\star(X^{CI}) \quad \text{almost everywhere.} \tag{6}$$

*Besides, under the assumptions of Lemma 4.1, the excess risk of chaining oracles compared to the Bayes risk $\mathcal{R}^\star$ is upper-bounded by:*

$$\mathcal{R}(f^\star \circ \Phi^{CI}) - \mathcal{R}^\star \leq \frac{1}{4}\mathbb{E}_M\left[\max\left(tr\left(\bar{H}^-_{mis,mis}\Sigma_{mis|obs,M}\right)^2, tr\left(\bar{H}^+_{mis,mis}\Sigma_{mis|obs,M}\right)^2\right)\right] \tag{7}$$

*Proof.*

$$Y - f^\star(X^{CI}) = (Y - \tilde{f}^\star(\widetilde{X})) + (\tilde{f}^\star(\widetilde{X}) - f^\star(X^{CI})) \tag{15}$$

$$\left(Y - f(X^{CI})\right)^2 = (Y - \tilde{f}^\star(\widetilde{X}))^2 + (\tilde{f}^\star(\widetilde{X}) - f^\star(X^{CI}))^2 \tag{16}$$

$$+ 2(Y - \tilde{f}^\star(\widetilde{X}))(\tilde{f}^\star(\widetilde{X}) - f^\star(X^{CI})) \tag{17}$$

$$= (Y - \tilde{f}^\star(\widetilde{X}))^2 + (\tilde{f}^\star(\widetilde{X}) - f^\star(X^{CI}))^2 \tag{18}$$

$$+ 2(f^\star(X) - \tilde{f}^\star(\widetilde{X}))(\tilde{f}^\star(\widetilde{X}) - f^\star(X^{CI})) \tag{19}$$

$$+ 2\epsilon(\tilde{f}^\star(\widetilde{X}) - f^\star(X^{CI})) \tag{20}$$

$$\mathbb{E}\left[\left(Y - f^\star(X^{CI})\right)^2\right] = \mathcal{R}^\star + \mathbb{E}\left[\left(\tilde{f}^\star(\widetilde{X}) - f^\star(X^{CI})\right)^2\right] \tag{21}$$

where we used the definition of the Bayes rate. Moreover, term (20) vanishes when taking the expectation w.r.t $\epsilon$ because $\mathbb{E}\left[\epsilon|X_{obs}, M\right] = 0$ and $\epsilon$ in uncorrelated with $X$ or $M$, and term (19) vanishes when taking the expectation w.r.t $X_{mis}|X_{obs}, M$ because by definition $\mathbb{E}_{X_{mis}|X_{obs},M}\left[f^\star(X_{obs}, X_{mis})\right] = \tilde{f}^\star(\widetilde{X})$.

Clearly, $f^\star \odot \Phi^{CI}$ is Bayes optimal if ans only if:

$$\mathbb{E}\left[\left(\tilde{f}^\star(\widetilde{X}) - f^\star(X^{CI})\right)^2\right] = 0 \tag{22}$$

$$\iff \sum_M \int P(X_{obs}, M)\left(\tilde{f}^\star(\widetilde{X}) - f^\star(X^{CI})\right)^2 dX_{obs} = 0 \tag{23}$$

$$\iff \forall M, X_{obs} : P(X_{obs}, M) > 0, \ \tilde{f}^\star(\widetilde{X}) = f^\star(X^{CI}) \quad \text{almost everywhere.} \tag{24}$$

where equality 24 is true since all terms are positive.

Besides, by Lemma 4.1, we have:

$$\frac{1}{2}tr\left(\bar{H}^-_{mis,mis}\Sigma_{mis|obs,M}\right) \leq \tilde{f}^\star(\widetilde{X}) - f^\star(X^{CI}) \leq \frac{1}{2}tr\left(\bar{H}^+_{mis,mis}\Sigma_{mis|obs,M}\right). \tag{25}$$

By convexity of the square function, it follows that:

$$\left(\tilde{f}^\star(\widetilde{X}) - f^\star(X^{CI})\right)^2 \leq \frac{1}{2}\max\left(tr\left(\bar{H}^-_{mis,mis}\Sigma_{mis|obs,M}\right)^2, tr\left(\bar{H}^+_{mis,mis}\Sigma_{mis|obs,M}\right)^2\right). \tag{26}$$

Finally, by taking the expectation on both sides:

$$\mathbb{E}\left[\left(\tilde{f}^\star(\widetilde{X}) - f^\star(X^{CI})\right)^2\right] \leq$$
$$\frac{1}{2}\mathbb{E}_M\left[\max\left(tr\left(\bar{H}^-_{mis,mis}\Sigma_{mis|obs,M}\right)^2, tr\left(\bar{H}^+_{mis,mis}\Sigma_{mis|obs,M}\right)^2\right)\right]. \tag{27}$$

Combining equation (21) with inequality (27) concludes the proof. $\qquad\square$

## A.8 Proof of Proposition 4.2

**Proposition 4.2** (Regression function discontinuities). *Suppose that $f^\star \circ \Phi^{CI}$ is not Bayes optimal, and that the probability of observing all variables is strictly positive, i.e., for all $x$, $P(M = (0,\ldots,0), X = x) > 0$. Then there is no continuous function $g$ such that $g \circ \Phi^{CI}$ is Bayes optimal.*

*Proof.* We will prove this result by contradiction. Suppose that (i) $f^\star \circ \Phi^{CI}$ is not Bayes optimal, (ii) the probability of observing all variables is strictly positive, (iii) there exists a function $g$ continuous such that $g \circ \Phi^{CI}$ is Bayes optimal.

Following a reasoning similar to the one in the proof of proposition 4.1, we can show that $g \circ \Phi^{CI}$ is Bayes optimal if and only if:

$$\forall M, X_{obs} : P(X_{obs}, M) > 0, \quad \mathbb{E}\left[f^\star(X)|X_{obs}, M\right] = g(X^{CI}) \quad \text{almost everywhere.}$$

In particular since for all $x$, the joint probability $P(M = (0,\ldots,0), X = x)$ of observing all variables is strictly positive, $g$ should satisfy this equality for $M = (0,\ldots,0)$, i.e.:

$$f^\star(X) = g(X) \quad \text{almost everywhere.}$$

Since $g$ is continuous, it implies $g = f^\star$. Since by assumption, $f^\star$ is not Bayes optimal, then $g$ is not either, which is a contradiction. $\qquad\square$

## A.9 Example of a case where no continuous corrected imputation exists.

Let:

$$f^\star : \mathbb{R}^2 \to \mathbb{R}$$
$$(X_1, X_2) \mapsto X_2^3 - 3X_2$$

and let:

$$X_2 = X_1 + \epsilon \quad \text{with} \quad \begin{aligned} &\mathbb{E}\left[\epsilon|X_1, M = (0,1)\right] = 0 \\ &\mathbb{E}\left[\epsilon^2|X_1, M = (0,1)\right] = \sigma^2, \sigma^2 > 1 \\ &\mathbb{E}\left[\epsilon^3|X_1, M = (0,1)\right] = 0 \end{aligned}$$

Suppose that $X_2$ is missing. Then the Bayes predictor is given by:

$$\begin{aligned}
\tilde{f}^\star(X_1, M = (0,1)) &= \mathbb{E}\left[f^\star(X)|X_1, M = (0,1)\right] \\
&= \mathbb{E}\left[X_2^3 - 3X_2|X_1, M = (0,1)\right] \\
&= \mathbb{E}\left[(X_1 + \epsilon)^3 - 3(X_1 + \epsilon)|X_1, M = (0,1)\right] \\
&= \mathbb{E}\left[X_1^3 + \epsilon^3 + 3X_1\epsilon^2 + 3X_1^2\epsilon - 3X_1 - 3\epsilon)|X_1, M = (0,1)\right] \\
&= X_1^3 + 3X_1(\sigma^2 - 1)
\end{aligned}$$

Clearly, the Bayes predictor for $M = (0,1)$ is:

- continuous,
- non-decreasing since $\sigma^2 > 1$,
- $\displaystyle\lim_{X1 \to +\infty} \tilde{f}^\star(X_1, M = (0,1)) = +\infty$ and $\displaystyle\lim_{X1 \to -\infty} \tilde{f}^\star(X_1, M = (0,1)) = -\infty$.

Proof by contradiction: Suppose that there exists a function $\Phi : \mathbb{R} \to \mathbb{R}$ (i) continuous and (ii) such that for all $X_1$, $f^\star(X_1, \Phi(X_1)) = \tilde{f}^\star(X_1, M = (0,1))$.

Let $x_1^+ \in \mathbb{R}$ such that $\tilde{f}^\star(X_1 = x_1^+, M = (0,1)) > 2$. $x_1^+$ exists since $\displaystyle\lim_{X1 \to +\infty} \tilde{f}^\star(X_1, M = (0,1)) = +\infty$. Clearly,

$$f^\star(x_1^+, X_2) = \tilde{f}^\star(x_1^+, M = (0,1)) \iff X_2 = x_2^+ \quad \text{with} \quad x_2^+ > 2.$$

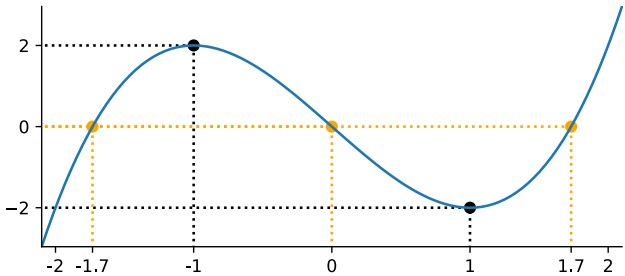

Figure 6: Graph of $X_2 \mapsto f^\star(X_1, X_2)$

Similarly, let $x_1^- \in \mathbb{R}$ such that $\tilde{f}^\star(X_1 = x_1^-, M = (0,1)) < -2$. $x_1^-$ exists since $\lim_{X1 \to -\infty} \tilde{f}^\star(X_1, M = (0,1)) = -\infty$. Clearly,

$$f^\star(x_1^-, X_2) = \tilde{f}^\star(x_1^-, M = (0,1)) \iff X_2 = x_2^- \quad \text{with} \quad x_2^- < -2.$$

So $\Phi$ must satisfy:

$$\Phi(x_1^-) = x_2^- < -2$$
$$\Phi(x_1^+) = x_2^+ > 2$$

Note that since the Bayes predictor is non-decreasing, we have $x_1^- < x_1^+$. Since $\Phi$ is continuous, there exists $\check{x}_1 \in [x_1^-, x_1^+]$ and $\hat{x}_1 \in [x_1^-, x_1^+]$ such that $\check{x}_1 < \hat{x}_1$ and $\Phi(\check{x}_1) = -1$ and $\Phi(\hat{x}_1) = 1$. It implies that:

$$f^\star(\check{x}_1, \Phi(\check{x}_1)) = f^\star(\check{x}_1, -1) = 2 > -2 = f^\star(\hat{x}_1, 1) = f^\star(\hat{x}_1, \Phi(\hat{x}_1)).$$

This implies that the function $X_1 \mapsto f^\star(X_1, \Phi(X_1))$ cannot be non-decreasing. Since the Bayes predictor is non-decreasing, the two cannot be equal. CONTRADICTION.

### A.10 Proof of Proposition 4.3

We start by proving the result for a given missing pattern $m \in \{0,1\}^d$. Take $r \in \{1, \ldots, d-1\}$ and consider a missing pattern $m$ such that $|obs(m)| = r$. We let $F : \mathbb{R}^r \times \mathbb{R}^{d-r} \to \mathbb{R}$ defined, for all $(x_{obs}, x_{mis})$ as

$$F(x_{obs}, x_{mis}) = f^\star(x_{obs}, x_{mis}) - \tilde{f}^\star(x_{obs}, m). \tag{28}$$

Our aim is to find, for all $x_{obs}$, a value $x_{mis}$ (depending continuously on $x_{obs}$) satisfying

$$F(x_{obs}, x_{mis}) = 0. \tag{29}$$

To this aim, we check the assumptions of Theorem 6 in Arutyunov and Zhukovskiy [2019] for the function $F$. The desired conclusion will follow.

Since $f^\star$ is uniformly continuous and twice continuously differentiable, condition $1-3$ of Theorem 6 in Arutyunov and Zhukovskiy [2019] are satisfied. To verify the next condition, we have to prove that there exists $(x_{obs,0}, x_{mis,0})$ such that $F(x_{obs,0}, x_{mis,0}) = 0$. Note that this is equivalent to finding $(x_{obs,0}, x_{mis,0})$ satisfying

$$f^\star(x_{obs,0}, x_{mis,0}) = \tilde{f}^\star(x_{obs,0}, m) = \mathbb{E}\left[f^\star(X)|X_{obs} = x_{obs,0}, M = m\right], \tag{30}$$

by definition of the regression function $\tilde{f}^\star$. By assumption, the support of $X_{mis}|X_{obs} = x_{obs,0}, M = m$ is connected. Therefore, the intermediate value theorem can be applied and proves the existence of a pair $(x_{obs,0}, x_{mis,0})$ satisfying equation (30). Finally, by assumption, the regularity condition (GR1) in Arutyunov and Zhukovskiy [2019] is satisfied. This proves that there exists a continuous mapping $\phi^{(m)} : \mathbb{R}^r \to \mathbb{R}^{d-r}$ such that

$$F(x_{obs}, \phi^{(m)}(x_{obs})) = 0. \tag{31}$$

The previous reasoning holds for all missing patterns $m$, such that $|mis(m)| \geq 1$. Besides the result is clear for $r = 0$ since the imputation function is reduced to a constant in this case (no components of $X$ are observed). On the contrary, in the case where all covariates are observed ($r = d$), no imputation function is needed. Therefore, the result holds for all $0 \leq r \leq d$, which concludes the proof.

## B  Additional results

### B.1  NeuMiss+MLP architecture

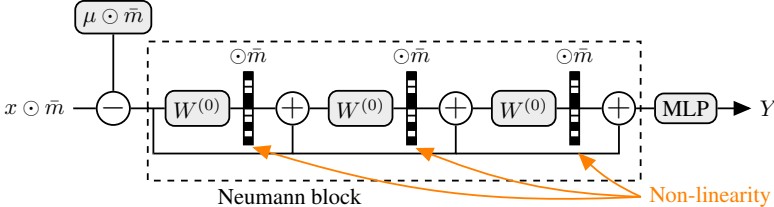

Figure 7: **(Non-linear) NeuMiss+MLP network architecture with a Neumann block of depth 3** — $\bar{m} = 1 - m$. MLP stands for a standard multi-layer perceptron with ReLU activations.

### B.2  Expressions of $f^\star_{bowl}$, $f^\star_{wave}$ and $f^\star_{break}$ and the corresponding Bayes predictors.

**Expressions of $f^\star_{bowl}$, $f^\star_{wave}$ and $f^\star_{break}$.**  The functions $f^\star$ used in the experimental study are defined as:

$$f^\star_{bowl}(X) = \left(\beta^\top X + \beta_0 - 1\right)^2$$

$$f^\star_{wave}(X) = (\beta^\top X + \beta_0 - 1) + \sum_{(a_i, b_i) \in S} a_i \, \Phi\left(\gamma\left(\beta^\top X + \beta_0 + b_i\right)\right)$$

$$f^\star_{break}(X) = \left(\beta^\top X + \beta_0\right) + 3 \times \mathbb{1}_{\beta^\top X + \beta_0 > 1}$$

where $\Phi$ the standard Gaussian cdf, $\gamma = 20\sqrt{\frac{\pi}{8}}$ and $S = \{(2, -0.8), (-4, -1), (2, -1.2)\}$. $\beta$ is chosen as a vector of ones rescaled so that $\mathrm{var}(\beta^\top X) = 1$. These functions are depicted in Figure 3.

**Expressions of the Bayes predictors.**  The expressions of the corresponding Bayes predictors are given by:

$$\tilde{f}^\star_{bowl}(\widetilde{X}) = \mathbb{E}\left[f^\star_{bowl}(X) | X_{obs}, M\right] \tag{32}$$

$$= \left(\beta^\top_{obs} X_{obs} + \beta^\top_{mis}\mu_{mis|obs,M} + \beta_0 - 1\right)^2 + \beta^\top_{mis}\Sigma_{mis|obs,M}\beta_{mis} \tag{33}$$

$$\tilde{f}^\star_{wave}(\widetilde{X}) = \mathbb{E}\left[f^\star_{wave}(X) | X_{obs}, M\right] \tag{34}$$

$$= \beta^\top_{obs} X_{obs} + \beta^\top_{mis}\mu_{mis|obs,M} + \beta_0 - 1 \tag{35}$$

$$+ \sum_{(a_i, b_i) \in S} a_i \, \Phi\left(\frac{\beta^\top_{obs} X_{obs} + \beta^\top_{mis}\mu_{mis|obs,M} + \beta_0 + b_i}{\sqrt{1/\gamma^2 + \beta^\top_{mis}\Sigma_{mis|obs,M}\beta_{mis}}}\right) \tag{36}$$

$$\tilde{f}^\star_{break}(\widetilde{X}) = \mathbb{E}\left[f^\star_{break}(X) | X_{obs}, M\right] \tag{37}$$

$$= \beta^\top_{obs} X_{obs} + \beta^\top_{mis}\mu_{mis|obs,M} + \beta_0 + 3\left(1 - \Phi\left(\frac{1 - \mu_{mis|obs,M}}{\beta^\top_{mis}\Sigma_{mis|obs,M}\beta_{mis}}\right)\right) \tag{38}$$

with $\mu_{mis|obs,M}$ and $\Sigma_{mis|obs,M}$ the mean and covariance matrix of the conditional distribution $P(X_{mis}|X_{obs}, M)$. Below, we give the expression of these parameters for the MCAR and Gaussian self-masking missing data mechanisms. Let $\mu_{mis|obs}$ and $\Sigma_{mis|obs}$ the mean and covariance matrix of the conditional distribution $P(X_{mis}|X_{obs})$. Since the data is generated according to a multivariate Gaussian distribution $\mathcal{N}(\mu, \Sigma)$, we have:

$$\mu_{mis|obs} = \mu_{mis} + \Sigma_{mis|obs}\Sigma^{-1}_{obs}(X_{obs} - \mu_{obs})$$

$$\Sigma_{mis|obs} = \Sigma_{mis,mis} - \Sigma_{mis,obs}\Sigma^{-1}_{obs}\Sigma_{obs,mis}$$

In the MCAR case, we simply have $\Sigma_{mis|obs,M} = \Sigma_{mis|obs}$ and $\mu_{mis|obs,M} = \mu_{mis|obs}$. In the Gaussian self-masking case, it has been shown in Le Morvan et al. [2020a] that $P(X_{mis}|X_{obs}, M)$ is

again Gaussian but with parameters:

$$\Sigma_{mis|obs,M} = \left( D_{mis,mis}^{-1} + \Sigma_{mis|obs}^{-1} \right)^{-1}$$

$$\mu_{mis|obs,M} = \Sigma_{mis|obs,M} \left( D_{mis,mis}^{-1} \widetilde{\mu}_{mis} + \Sigma_{mis|obs}^{-1} \mu_{mis|obs} \right)$$

where $\tilde{\mu}$ and $D$ are parameters of the Gaussian self-masking missing data mechanism. Finally, we detail below the derivations to obtain the expression of the Bayes predictors.

**Derivation of the Bayes predictor for $f_{bowl}^\star$.**

$$f_{bowl}^\star(X) = \left( \beta^\top X + \beta_0 - 1 \right)^2 \tag{39}$$

$$= \left( \beta_{obs}^\top X_{obs} + \beta_{mis}^\top X_{mis} + \beta_0 - 1 \right)^2 \tag{40}$$

$$= \left( \beta_{obs}^\top X_{obs} + \beta_{mis}^\top (X_{mis} - \mu_{mis|obs,M}) + \beta_{mis}^\top \mu_{mis|obs,M} + \beta_0 - 1 \right)^2 \tag{41}$$

$$= \left( \beta_{obs}^\top X_{obs} + \beta_{mis}^\top \mu_{mis|obs,M} + \beta_0 - 1 \right)^2 + \left( \beta_{mis}^\top (X_{mis} - \mu_{mis|obs,M}) \right)^2 \tag{42}$$

$$+ 2\beta_{mis}^\top (X_{mis} - \mu_{mis|obs,M}) \left( \beta_{obs}^\top X_{obs} + \beta_{mis}^\top \mu_{mis|obs,M} + \beta_0 - 1 \right) \tag{43}$$

Now taking the expectation with regards to $P(X_{mis}|X_{obs}, M)$, the last term vanishes and we get:

$$\mathbb{E}\left[f_{bowl}^\star(X)|X_{obs}, M\right] = \left( \beta_{obs}^\top X_{obs} + \beta_{mis}^\top \mu_{mis|obs,M} + \beta_0 - 1 \right)^2 + \beta_{mis}^\top \Sigma_{mis|obs,M} \beta_{mis} \tag{44}$$

**Derivation of the Bayes predictor for $f_{wave}^\star$.**

$$f_{wave}^\star(X) = (\beta^\top X + \beta_0 - 1) + \sum_{(a_i,b_i)\in S} a_i \, \Phi\left( \gamma \left( \beta^\top X + \beta_0 + b_i \right) \right) \tag{45}$$

$$= (\beta_{obs}^\top X_{obs} + \beta_{mis}^\top X_{mis} + \beta_0 - 1) \tag{46}$$

$$+ \sum_{(a_i,b_i)\in S} a_i \, \Phi\left( \gamma \left( \beta_{obs}^\top X_{obs} + \beta_{mis}^\top X_{mis} + \beta_0 + b_i \right) \right) \tag{47}$$

Define $T^{(m)} = \beta_{mis}^\top X_{mis}$. Since $P(X_{mis}|X_{obs}, M)$ is Gaussian in both the MCAR and Gaussian self-masking cases, $P(T^{(m)}|X_{obs}, M)$ is also Gaussian with mean and variance given by:

$$\mu_{T^{(m)}|X_{obs,M}} = \beta_{mis}^\top \mu_{mis|obs,M} \tag{48}$$

$$\sigma^2_{T^{(m)}|X_{obs,M}} = \beta_{mis}^\top \Sigma_{mis|obs,M} \beta_{mis} \tag{49}$$

To compute the Bayes predictor, we now need to compute the quantity:

$$\mathbb{E}_{T^{(m)}|X_{obs,M}} \left[ \Phi\left( \gamma \left( \beta_{obs}^\top X_{obs} + T^{(m)} + \beta_0 + b_i \right) \right) \right]$$

This expectation can then be computed following [Bishop, 2006] (section 4.5.2) which gives the result.

**Derivation of the Bayes predictor for $f_{break}^\star$.**

$$f_{break}^\star(X) = \left( \beta^\top X + \beta_0 \right) + 3 \times \mathbb{1}_{\beta^\top X + \beta_0 > 1} \tag{50}$$

$$\mathbb{E}\left[f_{break}^\star(X)|X_{obs}, M\right] = \beta_{obs}^\top X_{obs} + \beta_{mis}^\top \mu_{mis|obs,M} + \beta_0 \tag{51}$$

$$+ 3 \times \int P(X_{mis}|X_{obs}, M) \mathbb{1}_{\beta_{obs}^\top X_{obs} + \beta_{mis}^\top X_{mis} + \beta_0 > 1} dX_{mis} \tag{52}$$

Let $U^{(m)} = \beta_{obs} X_{obs} + \beta_{mis} X_{mis} + \beta_0$. Since $P(X_{mis}|X_{obs}, M)$ is Gaussian in both the MCAR and Gaussian self-masking cases, $P(U^{(m)}|X_{obs}, M)$ is also Gaussian with mean and variance given by:

$$\mu_{U^{(m)}|X_{obs,M}} = \beta_{obs}^\top X_{obs} + \beta_{mis}^\top \mu_{mis|obs,M} + \beta_0 \tag{53}$$

$$\sigma^2_{U^{(m)}|X_{obs,M}} = \beta_{mis}^\top \Sigma_{mis|obs,M} \beta_{mis} \tag{54}$$

Using the law of the unconscious statistician, we get:

$$\mathbb{E}\left[f_{break}^{\star}(X)|X_{obs}, M\right] = \beta_{obs}^{\top}X_{obs} + \beta_{mis}^{\top}\mu_{mis|obs,M} + \beta_0 \tag{55}$$

$$+ 3 \times \int P(U^{(m)}|X_{obs}, M)\mathbb{1}_{U^{(m)}>1}dU^{(m)} \tag{56}$$

$$= \beta_{obs}^{\top}X_{obs} + \beta_{mis}^{\top}\mu_{mis|obs,M} + \beta_0 \tag{57}$$

$$+ 3 \times \left[1 - \mathbb{P}\left(U^{(m)} \le 1|X_{obs}, M\right)\right] \tag{58}$$

$$= \beta_{obs}^{\top}X_{obs} + \beta_{mis}^{\top}\mu_{mis|obs,M} + \beta_0 \tag{59}$$

$$+ 3 \times \left[1 - \Phi_{U^{(m)}|X_{obs},M}(1)\right] \tag{60}$$

## B.3 Supplementary experiments with $f_{break}^*$.

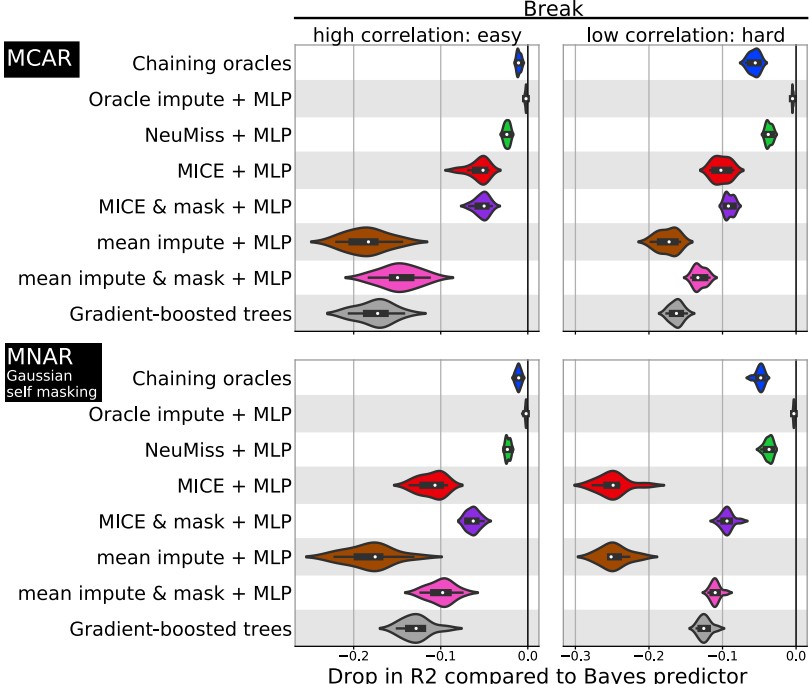

Figure 8: Performances (R2 score on a test set) compared to that of the Bayes predictor across 10 repeated experiments.