# OpenReview forum: "What’s a good imputation to predict with missing values?"
_NeurIPS.cc/2021/Conference — NeurIPS 2021 Spotlight_

### Official Review · Reviewer_1UHV · 2021-07-13

**Rating:** 7
**Confidence:** 3

**Summary:**

The authors look at the impute and regress framework and provide arguments as to Bayes optimality of the process for missing data mechanisms and imputation functions. It gives a general theoretical framework for the impute and regress procedures. Unlike the missing at random setups, they look at more general missing value mechanisms. They also argue that the imputation and regression processes should be properly aligned with each other and also provide a method for optimizing the two processes jointly.

**Limitations And Societal Impact:**

The authors do state that the choice of imputation function impacts the regression step and how the continuity and smoothness are not guaranteed making it hard to approximate. However, as stated in the previous section, they look at this case with the imputation being conditional expectation and also show the existence of continuous imputation functions that lead to Bayes optimality.

**Main Review:**

The paper is well written and provides a well-structured overview of the results. They provide a framework that gives a theoretical background for the impute-then-regress process and they look at a more general setup that does not require any structured missing data mechanisms such as MAR and etc. The authors show that for almost all infinitely differentiable, smooth impute functions, there is an impute-then-regress process that is Bayes optimal. They provide a constructive argument to that effect. The paper also looks at the case of using conditional expectations for the imputation and how this could result in regression functions with discontinuity. The authors also provide a chaining of NeuMiss and multi-layer perceptron to compute the impute-then-regress process. All the claims are clearly stated and properly presented and the math seems to hold as well.

**Time Spent Reviewing:**

2.5

---

> ### Author Response · Authors · 2021-08-09
> **Answers to the reviewer's questions and suggestions**
>
> We thank the reviewer for its time and useful comments.
>
> In what follows, we quote the reviewer’s questions or suggestions and provide responses.
>
>     Quote 1: “The authors do state that the choice of imputation function impacts the regression step and how the continuity and smoothness are not guaranteed making it hard to approximate. However, as stated in the previous section, they look at this case with the imputation being conditional expectation and also show the existence of continuous imputation functions that lead to Bayes optimality.”
>
> About this paragraph ‘Limitations and Societal Impact’, we agree with the reviewer, although we see this limitation about a possibly hard regression step more as a result of our theoretical work on Impute-then-Regress methods that we hope will inspire future developments.
>
> Indeed, our theoretical results are asymptotic and for some choices of imputation function, the regression step may be hard. Thus, as stated by the reviewer, we start by analysing what happens when imputing by the oracle imputation. This is a natural choice to start with since it is what is targeted by most imputation methods. We show that it is a good choice (although not optimal) if some conditions are met and follow up by showing the existence of continuous imputations that lead to Bayes optimality. An open question is how hard it is to approximate such Bayes optimal solution, be it by joint optimization or other strategies to define.

---

### Official Review · Reviewer_wpgv · 2021-07-15

**Rating:** 8
**Confidence:** 4

**Summary:**

The paper first seeks to study the theoretical properties of impute-then-regress procedures -- which are widely used. They show that most any imputations admit Bayes consistent estimators; however, that estimation may be difficult in certain settings. They show that conditional imputation is not necessarily an optimal strategy, supply bounds on the associated risk, and discuss when it can be expected to perform reasonably well. In particular, conditional imputation under some conditions requires learning a discontinuous regression function for optimal performance. Given all this, they examine the conditions under which imputations exist such that impute-then-regress procedures are optimal. Lastly, they propose a procedure for joint optimization of an imputation and a regression scheme that is empirically shown to 1. outperform common impute-then-regression procedures as well as gradient boosting which naturally can accommodate missingness and 2. perform similarly to oracle estimators in certain settings.

**Limitations And Societal Impact:**

Yes.

**Main Review:**

**Originality**: I was not familiar with general work on the validity of impute-then-regress schemes, though this submission seems to clearly move beyond the restrictive settings considered in past papers. In general, there is not much related work cited on this concrete task (not that I am sure more exists). The actual method proposed to take advantage of the demonstrated theory -- NeuMiss+MLP -- is not very innovative, though it is not meant to be; the point is simply that joint optimization of how to impute and how to regress is superior. The proof technique of Theorem 3.1. seems to be original, though I did not examine it carefully.

**Quality**: Claims are backed up by theoretical results, the proofs of which I did not examine closely. There is empirical support for the overall argument of the paper. Thought it is not the focus, it might have been good to touch upon the limitations of NeuMiss. I would also have liked to see a discontinuous $f^*$, though I understand that the Bayes predictor might be harder to compute.

**Clarity**: I thought the paper read very well. A few typos to address before CR, but I like the overall structure.

**Significance**:  I don't think the prevalence of "first impute, then regress" as a strategy for handling missing data can be understated and for this reason I find the work very compelling. It is possible that in the future the standard shifts towards a joint treatment of the two tasks and this paper provides evidence for why that would be beneficial. It would have been useful to comment on joint schemes beyond the one proposed here. For example, do researchers that don't want to rely on an uninterpretable method for regression have any hope of adopting / adapting the approach here?

**Time Spent Reviewing:**

2

---

> ### Author Response · Authors · 2021-08-09
> **Answers to the reviewer's questions and suggestions**
>
> We thank the reviewer for its time and useful comments.
>
> In what follows, we quote the reviewer’s questions or suggestions and provide responses.
>
>
>  	Quote 1 - “In general, there is not much related work cited on this concrete task (not that I am sure more exists)”
> It is true that while Impute-then-Regress strategies are heavily used in practice, there are almost no theoretical works on the subject as far as we know.
>
>  	Quote 2 - “The proof technique of Theorem 3.1. seems to be original, though I did not examine it carefully.”
> The proof technique relies on theorems from differential topology, seeing the data after imputation as a collection of manifolds. We are not aware of proofs that make use of similar arguments, and therefore think that it makes the proof technique original.
>
>  	Quote 3 - “it might have been good to touch upon the limitations of NeuMiss”
> This is true we will add a comment. The main limitation of NeuMiss is that the theory that backs this architecture holds under the hypothesis of Gaussian data. Experiments to test the robustness of the architecture to non-Gaussian data are ongoing, but up to now promising.
>
> 	Quote 4 - “I would also have liked to see a discontinuous $f^\star$, though I understand that the Bayes predictor might be harder to compute.”
> We did the experiment with $f^\star = \beta^Tx + 3 \mathbb 1 (\beta^Tx > 1)$. On the halfspace $\beta^Tx < 1$ it is a linear function with no offset and on the other halfspace, it is a linear function with an offset, which thus creates a discontinuity. We ran the experiments in MCAR.
>
> We do not have an analytical expression for the Bayes predictor in this case, so we cannot compare the performances of the methods with regards to the Bayes predictor.
>
> However, in terms of relative performances among the methods, the results are very similar to the ones with the wave and bowl functions, except now the performance of the chained oracle in low correlation (hard) setting is below NeuMiss but above MICEMLP.
>
> We will add this experiment to the manuscript.
>
> 	Quote 5 -  “It would have been useful to comment on joint schemes beyond the one proposed here.”
> There has been up to now few works on joint schemes, which is linked to the fact that there are many works on imputation but few works on supervised supervised learning with missing values.
>
> However, we can see Missing Incorporated Attribute strategy (or MIA) as one example. MIA is a technique to directly handle missing values without imputation in tree-based methods, which we use in our experiments with GBRT. But it can be shown that it corresponds to doing an imputation implicitly that is learned to predict as well as possible.
>
> 	Quote 6 - “For example, do researchers that don't want to rely on an uninterpretable method for regression have any hope of adopting / adapting the approach here?”
> If one wants to do a linear regression with missing values in order to interpret the parameters, it seems a reasonable idea to derive the EM algorithm that will give the maximum likelihood estimator with missing values. However, this approach is limited by the fact that it is only valid in Missing At Random settings, unless a missing data mechanism is specified.
>
> Here our goal is to predict at best despite missing values, so the learned imputation may or may not be close to the oracle imputation depending on the problem at hand. But one can still use a block like NeuMiss that learns an imputation chained with a linear model.
>
> Otherwise, one notable advantage of having a fully differentiable architecture such as NeuMiss+MLP is that recent tools from explainable AI developments can be applied.

---

### Official Review · Reviewer_Jo6j · 2021-07-16

**Rating:** 9
**Confidence:** 3

**Summary:**

In handling missing data, a standard approach is to simply impute and then run a prediction model that requires fully-revealed data. However, this standard "impute-then-regress" approach is largely a hack without theoretical grounding. This paper provides theory to justify this standard approach, with a surprising finding that it turns out that for almost all imputation functions, impute-then-regress with a sufficiently powerful learner is Bayes optimal. Moreover, the paper theoretically justifies why imputation and regression should be done jointly rather than separately as doing so separately can require learning discontinuous functions that are harder to learn. Numerical experiments corroborate the theoretical findings.

**Limitations And Societal Impact:**

The authors have provided a sufficient answer (see checklist 1(c)).

**Main Review:**

Overall, this paper is very well-written and I find it to constitute a major advance in our understanding of the standard impute-then-regress approach (which shows up in a massive number of applied papers throughout scientific literature but lacks theoretical grounding). The theoretical findings (summarized in my summary above) are surprising and informative of the direction researchers should shift toward in handling missing data.

Strengths:
- extremely well-motivated paper
- compelling theoretical and experimental results
- detailed numerical experiments

Weaknesses:
- especially as the claim of Theorem 3.1 is surprising regarding the "almost all" imputation methods, perhaps it would be instructive providing an example problem setup and imputation method that does not lead to Bayes optimality and relating such an example to what is done in practice (perhaps such an example is contrived and basically all practical imputation methods really do fall under the coverage of Theorem 3.1)

**Time Spent Reviewing:**

2

---

> ### Author Response · Authors · 2021-08-09
> **Answers to the reviewer's questions and suggestions**
>
> We thank the reviewer for its time and useful comments.
>
> In what follows, we quote the reviewer’s questions or suggestions and provide responses.
>
> 	Quote 1 - “it would be instructive providing an example problem setup and imputation method that does not lead to Bayes optimality and relating such an example to what is done in practice (perhaps such an example is contrived and basically all practical imputation methods really do fall under the coverage of Theorem 3.1)”
>
> Such an example is indeed contrived and as the reviewer says, basically all practical imputation methods will fall under the coverage of Theorem 3.1. However, because the Theorem is true for **almost all** imputation functions and not **all** of them, we can build an example with a particular choice of imputation function that does not lead to Bayes optimality.
>
> Suppose you have Gaussian data in 2D. Let $a \in \mathbb R$. Suppose that if $x_2$ is missing, you impute it as a*$x_1$. And suppose that if $x_1$ is missing, you impute it as 1/a*$x_2$. Then the manifolds on which the data with either $x_1$ missing or $x_2$ missing are projected are exactly the same (the same line in the 2D space). This is a particular case where the two manifolds won’t be transverse, and thus where the Theorem does not hold.
>
> However as shown in the proof with the Thom transversality theorem, almost all imputation functions will lead to transverse manifolds.
>
> We will add this example in the manuscript or Appendix.

---

### Official Review · Reviewer_YYbQ · 2021-07-19

**Rating:** 7
**Confidence:** 3

**Summary:**

This paper proposes to jointly optimize imputation and regression to learn from data with missing values, backed by theoretical analysis of impute-then-regress procedures. In particular, the authors show that the impute-then-regress procedure is asymptotically Bayes optimal for almost all (smooth) imputation functions, regardless of the missing data mechanism (including MNAR). However, apart from restrictive special cases, learning the regression function for conditional imputation to obtain a Bayes optimal function is hard due to discontinuities; similar with correcting the imputation for a given oracle function. The authors thus propose to jointly optimize impute-then-regress procedures, in particular chaining a NeuMiss architecture with an MLP. The proposed approach is empirically evaluated on synthetic data with various combinations of imputation and regression methods.

**Limitations And Societal Impact:**

Yes

**Main Review:**

The paper considers an important problem of learning with missing data, under a general setting with little assumptions on the oracle function or the missingness mechanism, and thus can be relevant in many practical scenarios.

It is also well-written overall. One suggestion for clarity is to formally define “almost all” in Theorem 3.1 sooner.

This work contains interesting and novel theoretical results. In particular, I was surprised that impute-then-regress is Bayes optimal even under MNAR for almost all imputation functions, including a simple constant function. I believe Proposition 4.1 is also an important result, because many existing works on imputation implicitly or explicitly target to learn the conditional imputation and oracle labeling function.

However, I was less convinced by the conclusion to jointly optimize imputation and regression. According to the authors, for a fixed imputation function, the Bayes optimal regression function may not be continuous, and vice versa. It was not clear how joint optimization addresses this, and whether the joint learning could be confused. The specific choice of NeuMiss was also not very well motivated.

Alternatively, if the goal is not to learn the “best” imputation and “best” regression functions but rather to chain two architectures to optimize predictive performance, how does it compare to simply optimizing one architecture that takes as input the given observations concatenated with missingness information?

Another recommendation is to also evaluate on real-world benchmark dataset to compare against state-of-the-art learning from missing data.

**Time Spent Reviewing:**

4 hours

---

> ### Author Response · Authors · 2021-08-09
> **Answers to the reviewer's questions and suggestions**
>
> We thank the reviewer for its time and useful comments.
>
> In what follows, we quote the reviewer’s questions or suggestions and provide responses.
>
>
> 	Quote 1 - “However, I was less convinced by the conclusion to jointly optimize imputation and regression. According to the authors, for a fixed imputation function, the Bayes optimal regression function may not be continuous, and vice versa. It was not clear how joint optimization addresses this, and whether the joint learning could be confused. The specific choice of NeuMiss was also not very well motivated.”
>
> We understand the point of the reviewer. Indeed, a joint optimization of the imputation and regression functions does not necessarily converge to an optimal solution where both functions are smooth (it is not even always possible as we show in the theoretical par 4.3).
>
> We will try to make clear why we used a joint optimization of imputation and regression with NeuMiss. After theorem 3.1 on Bayes optimality of Impute-then-Regress, we note that some imputations can lead to hard regression problems and follow up by:
>   * studying the case of an oracle imputation. It is a natural point to start with since it is what most imputation methods target. We show it is a good choice if some conditions are met although it is not optimal.
>   * We then show that continuous imputations that lead to Bayes optimality exist in some cases. However, how to approximate such functions in practice is a difficult open question.
>
> Thus, for the experimental part we resort to using a joint optimization scheme of imputation and regression with the hope that the learned imputation adapts to the regression task. The choice of using NeuMiss to model the imputation function is key and motivated by two main reasons:
>   * There exists values for the weights of NeuMiss so that it outputs an approximation of the oracle imputation (in the hypothesis of a Gaussian distribution) with an error that decays exponentially fast with its depth. This property is desirable in some cases as explained in Prop. 4.1. The actual learned weights of the NeuMiss layer may or may not correspond to the ones that yield an oracle imputation depending on what provides the best performance in regression.
>   * NeuMiss is a differentiable imputation layer, which allows it to be chained with a MLP and learned jointly with the regression function.
>
> This approach provides notable benefits in Missing Not at Random settings, where as can be seen from the results, the imputation function learned by NeuMiss is successfully adapted to reach better performances compared to using MICE imputation followed by MLP for example.
>
> Based on the comment of the reviewer, we will make clearer the motivation for using NeuMiss in part 5.
>
>
> 	Quote 2 - “Alternatively, if the goal is not to learn the “best” imputation and “best” regression functions but rather to chain two architectures to optimize predictive performance, how does it compare to simply optimizing one architecture that takes as input the given observations concatenated with missingness information?”
>
> One answer is given by “mean impute & mask + MLP”, which corresponds to imputing by the mean, concatenating with the missingness information, and fitting a feedforward fully connected MLP to this data. Note that we need to fill in the missing values (here simply by a constant, the mean) to be able to feed it to the MLP. Otherwise, one would need to resort to new architectures that can deal with arbitrary subsets of variables. It is an interesting path to explore, but it is unclear whether such architectures would provide better performances.
>
> This question echoes the one on the motivation for using NeuMiss. NeuMiss is an architecture that is designed to handle missing values, with the specificity that it uses the element-wise multiplication by the missingness indicator as non-linearity instead of more classical nonlinearities such as ReLU. As mentioned above, it has been shown theoretically that this architecture allows NeuMiss to efficiently compute (with a number of weights that is not too large) an oracle imputation if it is relevant for the regression problem, which is not possible with a classical MLP. Our experiments show that using NeuMiss is better than using a MLP, even in MNAR, where we recall that NeuMiss does not take the missingness indicator as input contrary to the MLP.

---

### Author Response · Authors · 2021-09-01
**General response**

Dear reviewers and area chair,

Thank you again for your constructive comments. We feel that all reviewers agree that this work is important (Impute-then-regress procedures are very common but little studied), and that it comes with strong and novel theoretical results as well as experimental evidence. We have addressed thoroughly the reviewers’ questions and we hope that our replies were clear and insightful. The most important additional factual elements that we have brought are:

 * To improve the reader’s insight into Theorem 3.1 (which states that Impute-then-regress procedures are Bayes consistent for *almost all* imputation functions), we have given a simple example with a particular choice of imputation function that does not lead to Bayes optimality. Indeed, our theorem is very general with a broad impact, and can be surprising (reviewers Jo6j and YYbQ).
 * To illustrate more our theoretical results, we will add the new simulations where $f^\star$ is a discontinuous function (see reply to reviewer wpgv).

Should any more clarification be useful, we will be happy to further discuss it with you. Thank you again for your time and consideration!

The authors

---

### Decision · Program_Chairs · 2021-09-27

**Decision:**

Accept (Spotlight)

**Comment:**

The paper studies prediction in the presence of missing values. A common approach to solving this problem is "impute-then-regress", independently imputing missing values and predicting using the imputed data. The authors show that this strategy is almost always Bayes optimal w.r.t. risk, but that a poor imputation may lead to learning a complex prediction function. Several strategies are evaluated and compared empirically on synthetic data.

All reviewers recognised the value of this work and pointed out its strong motivation and clarity in exposition.